# Single dose of chimeric dengue-2/Zika vaccine candidate protects mice and non-human primates against Zika virus

Whitney R. Baldwin[1,2], Holli A. Giebler[1,2], Janae L. Stovall[1], Ginger Young [2], Kelly J. Bohning[2], Hansi J. Dean[2], Jill A. Livengood[1,2] & Claire Y.-H. Huang [1✉]

The development of a safe and effective Zika virus (ZIKV) vaccine has become a global health priority since the widespread epidemic in 2015-2016. Based on previous experience in using the well-characterized and clinically proven dengue virus serotype-2 (DENV-2) PDK-53 vaccine backbone for live-attenuated chimeric flavivirus vaccine development, we developed chimeric DENV-2/ZIKV vaccine candidates optimized for growth and genetic stability in Vero cells. These vaccine candidates retain all previously characterized attenuation phenotypes of the PDK-53 vaccine virus, including attenuation of neurovirulence for 1-day-old CD-1 mice, absence of virulence in interferon receptor-deficient mice, and lack of transmissibility in the main mosquito vectors. A single DENV-2/ZIKV dose provides protection against ZIKV challenge in mice and rhesus macaques. Overall, these data indicate that the ZIKV live-attenuated vaccine candidates are safe, immunogenic and effective at preventing ZIKV infection in multiple animal models, warranting continued development.

[1] Arboviral Diseases Branch, Division of Vector-Borne Diseases, Centers for Disease Control and Prevention, Fort Collins, CO, USA. [2] Takeda Vaccines Inc., Cambridge, MA, USA. ✉email: yxh0@cdc.gov

Although historically recognized as a sporadic and mild arboviral infection resulting in limited human disease, Zika virus (ZIKV) recently spread to multiple countries and caused microcephaly and other congenital malformations in infants born to women infected during pregnancy[1]. ZIKV infections are typically asymptomatic, or mildly symptomatic with self-limiting acute febrile illness accompanied by rash, joint pain, myalgia, and/or conjunctivitis. However, severe illness outcomes including Guillain–Barré syndrome in adults and congenital ZIKV syndrome in newborns have caused major public health burdens[2]. Despite a significant decline in reported ZIKV cases after the 2015–2016 outbreak in the Americas, ZIKV transmission is expected to spread to new geographical locations due to climate change, increased urbanization and travel, and expansion of mosquito vectors[3]. Therefore, an important public health need exists for a safe and efficacious vaccine capable of mitigating the effects of future ZIKV epidemics. Numerous vaccine candidates have been developed in recent years[4], including a purified inactivated ZIKV vaccine[5,6] that we developed and is currently being evaluated in clinical trials[7]. Herein, we report the successful development of a live-attenuated vaccine (LAV) against ZIKV. A LAV has the potential to provide life-long immunity with fewer immunizations, which would be especially important for populations living in ZIKV endemic countries. Ultimately, multiple types of vaccine will provide comprehensive options to protect diverse human populations from the devastating effects of ZIKV infection.

Belonging to the *Flaviviradae* family and flavivirus genus, ZIKV and dengue viruses (DENVs) have a positive-sense RNA genome that contains 5'- and 3'-terminal untranslated regions (5'UTR and 3'UTR) and encodes a polyprotein that is processed into 3 structural proteins, capsid (C), premembrane (prM) and envelope (E), and 7 nonstructural proteins (NS1, NS2A, NS2B, NS3, NS4A, NS4B and NS5). The common genome organization of flaviviruses permits generation of chimeric viruses by interchanging the prM-E genes between 2 heterologous flaviviruses[8]. The ZIKV LAV candidates reported in this study are based on the chimeric DENV-2 PDK-53 vaccine platform that we previously developed for several flavivirus LAVs, including a tetravalent DENV vaccine (TDV) and a West Nile virus (WNV) vaccine (D2/WNV)[8,9]. The DENV-2 PDK-53 vaccine was originally generated by attenuation through serial cell passaging of wild-type (wt) DENV-2 16681 at Mahidol University (Thailand) and was provided to the Division of Vector-Borne Diseases, CDC for recombinant TDV development. Infectious cDNA clones of 16681 and PDK-53 strains were generated to identify the molecular determinants of PDK-53 attenuation[10]. The two viruses exhibit 9 genetic differences, 3 of which encode dominant attenuation determinants: 5'UTR-c57t, NS1-G53D, and NS3-E250V[11]. The DENV-2 PDK-53 vaccine was shown to be safe and immunogenic in early human trials[12]. TDV (Takeda's TAK-003, previously designated as DENVax) based on the chimeric DENV-2 PDK-53 platform is currently being evaluated in Phase 3 clinical trials and has demonstrated up to 80.7% efficacy against virologically-confirmed DENV[13,14]. Based on our experience with TDV and D2/WNV vaccine development, we extended the platform to the development of chimeric DENV-2/ZIKV (D2/ZK) LAV candidates. Here, we describe the engineering and preclinical evaluation of chimeric D2/ZK LAV candidates that replicate to high yield in a vaccine production cell line, are genetically stable, and retain all previously characterized DENV-2 PDK-53 phenotypic markers of attenuation. A single dose of the vaccine protects mice and non-human primates (NHPs) against ZIKV challenge.

## Results

**Derivation of D2/ZK vaccine candidate viruses.** Following the similar construct design for chimeric D2/WNV[8], we generated recombinant cDNA clones containing the prM-E genes of a contemporary ZIKV isolate in the DENV-2 PDK-53 vaccine or parental 16681 genetic backbone to derive chimeric D2/ZK-V or D2/ZK-P viruses, respectively (Fig. 1a). The D2/ZK-P viruses served as controls to assess the contributions of the PDK-53 determinants to attenuation of the chimeric virus.

As the Vero cell line from the WHO reference cell bank (RCB)[15] has been used to generate cGMP-certified cells for manufacturing multiple virus vaccines, including TAK-003, we aimed to develop a ZIKV LAV that could grow efficiently in this cell line. Initially, both chimeric constructs (designated D2/ZK-P0 and -V0) failed to generate infectious virus from RNA-transfected Vero cells, indicating chimerization of the 2 viral genomes was not fully compatible with viral replication in mammalian cells. In mosquito C6/36 cells, viable D2/ZK-P0 virus was recovered at a low titer, while D2/ZK-V0 virus was still not viable. This result was not surprising as PDK-53 is restricted for growth in C6/36 cells[8,16,17]. To encourage adaptation of the chimeric viruses in Vero cells, we conducted three-serial Vero passages of the D2/ZK-P0 virus that was recovered from transfected C6/36 cells. Virus culture after a single Vero cell passage resulted in significant adaptation. While the initial harvest from transfected C6/36 cells failed to produce detectable plaques in Vero cell monolayers, a single Vero cell passage resulted in a titer greater than $10^6$ plaque forming units (pfu)/mL in Vero monolayers. A mixed plaque phenotype (small to large) was observed from all Vero cell passages, with passage-3 virus producing the highest proportion of larger plaques. Upon plaque purification of the larger plaques of D2/ZK-P0 passage-3 virus, genome sequencing revealed 3 amino acid (AA) substitutions (E-Q465R, E-I484T, and E-I493F) that appeared in multiple plaque isolates. Introduction of E-Q465R and E-I484T into the D2/ZK-V0 construct (designated D2/ZK-V2) resulted in viable virus recovery from Vero cells, indicating at least one of the substitutions was critical for Vero cell adaptation (Fig. 1b).

Although the D2/ZK-V2 virus replicated well, a mixed plaque phenotype was observed after a single round of amplification in Vero cells, suggesting the virus was still evolving. Genome sequencing of multiple plaque isolates of the D2/ZK-V2 virus identified a total of 6 AA substitutions in E, NS1, NS2A, and NS4A. Among them, E-I493F, NS2A-K99N and NS4A-D23N appeared to be most significant for Vero cell adaptation, based on their higher frequency of occurrence and association with isolates producing uniform plaques. Interestingly, E-I493F, previously identified but not included in the D2/ZK-V2 construct, appeared again. As such, six additional chimeric clones containing various combinations of the substitutions at E-465, E-484, E-493, NS2A-99 and NS4A-23 were generated to investigate their relative significance in Vero cell adaptation. We determined E-Q465R and E-I493F to be the most critical substitutions, as chimeric constructs lacking either substitution resulted in rapid evolution of the virus to regain that specific substitution within a single Vero cell passage. We found NS2A-K99N and NS4A-D23N also contributed to Vero cell adaptation, but could not rule out potential contribution of E-I484T. Accordingly, D2/ZK-V4 and D2/ZK-V5 viruses containing 4 and 5 of the substitutions, respectively, were selected as potential vaccine candidates for evaluation (Fig. 1b).

Following a report indicating that prM-S17N (polyprotein S139N), having emerged in the contemporary Asian lineage of ZIKV, was correlated with increased infectivity in human and

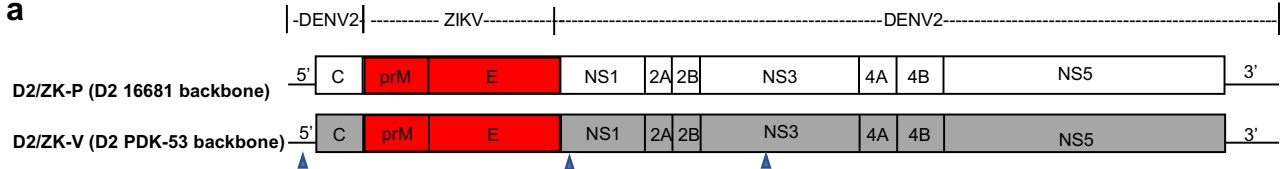

**Fig. 1 Genomic organization and modification of chimeric D2/ZK viruses. a** Genomic map of the D2/ZK viruses with prM-E of ZIKV SPH in the genomic background of either DENV-2 16681 (D2/ZK-P, parental) or DENV-2 PDK-53 (D2/ZK-V, vaccine). Blue triangles denote the 3 primary PDK-53 attenuation loci, 5-NCR c57t, NS1-G53D and NS3-E250V. **b** Genetic substitutions and plaque images of several D2/ZK-V viruses. Engineered mutations and a representative image of plaque morphology in Vero cells are indicated for each virus.

mouse neural progenitor cells, as well as more severe fetal microcephaly and mortality in mice[18], we also added prM-N17S substitution in the D2/ZK-V5 construct to generate a third candidate, D2/ZK-V5-Pr (Fig. 1b). Parental chimeric counterpart viruses (D2/ZK-P4, -P5, and P5-Pr) were generated as controls for phenotype characterization.

**Growth kinetics and genetic stability of D2/ZK-V candidates in Vero cells**. We evaluated the growth kinetics of the vaccine viruses using the WHO Vero RCB 10-87 cell line and medium lacking fetal bovine serum (FBS) to mimic a manufacturing process that could reduce the introduction of adventitious agents. All D2/ZK-V viruses replicated to a peak titer of approximately $10^7$ pfu/mL between days 4 and 8 post-infection (Fig. 2a). Growth of the D2/ZK-V viruses was similar to their D2/ZK-P counterparts, indicating the attenuated vaccine backbone did not significantly restrict growth of the chimeric viruses in Vero cells as previously observed[17]. All D2/ZK-V viruses replicated at a slower rate than ZIKV PRVABC59, but at similar or somewhat faster rates than DENV-2 16681. Interestingly, replication of both V5-Pr and P5-Pr viruses containing the prM-N17S were slightly delayed early in infection, but ultimately achieved a peak titer similar to the other D2/ZK viruses by day 8.

Multiple rounds of cell culture amplification of vaccine virus are required during large scale manufacturing. Therefore, the genetic stabilities of the candidate vaccine viruses were analyzed using next-generation sequencing (NGS) after 1, 5, and 10 serial Vero cell passages (P1, P5, P10) of the D2/ZK-V viruses (Fig. 2b). All engineered substitutions (Vero-adaptation and prM-N17S)

were retained in both duplicate passage lineages (Flask A and B) of the 3 candidate viruses. At P1, D2/ZK-V4 and -V5 retained genomic sequences nearly identical to the engineered cDNA clones with only 9-10 single nucleotide variants (SNVs) detected, while D2/ZK-V5-Pr had significantly more (39) SNVs. However, all P1 SNVs of the 3 candidate viruses were of very low frequency (< 5%) and most of the SNVs were not carried through to P5 or P10 of the same lineage. In the P5 and P10 viruses, the average number of SNVs present at ≥ 10% frequency was 5.3 (range of 3–8) and 9.8 (range of 6–15), respectively (Fig. 2b; Supplementary Tables 1–3). Of these SNVs, the number of AA changes was 3.4 (range of 2–5) and 5.0 (range of 2–9) for P5 and P10 viruses, respectively. Some of the SNVs in P5 were undetectable or at a lower frequency in P10 of the same lineage, suggesting these variants were unlikely to be the result of positive-selection pressure during cell passaging (Fig. 2b and Supplementary Tables 1–3). To identify potential SNVs arising from positive-selection pressure during cell passaging, we focused on the SNVs in each sample with ≥ 25% frequency in P10 (Table 1). In each P10 sample, 2–3 SNVs resulted in non-synonymous AA changes, mostly occurring in a single lineage, with the exception of the SNVs at prM-29, M-14, E-195, and NS4B-238 which each occurred in 2 of the 6 P10 viruses. A common SNV at 3'UTR-10649 was identified in 5 of the 6 P10 samples at a frequency around 30–40%. Most importantly, the three primary PDK-53 attenuation determinants were highly stable in both D2/ZK-V4 and -V5 virus without any detectable reversion or SNV (Fig. 2b). In contrast, both P10 samples of the D2/ZK-V5-Pr virus showed some degree of reversion at these loci. P10-A showed 34% and 3%

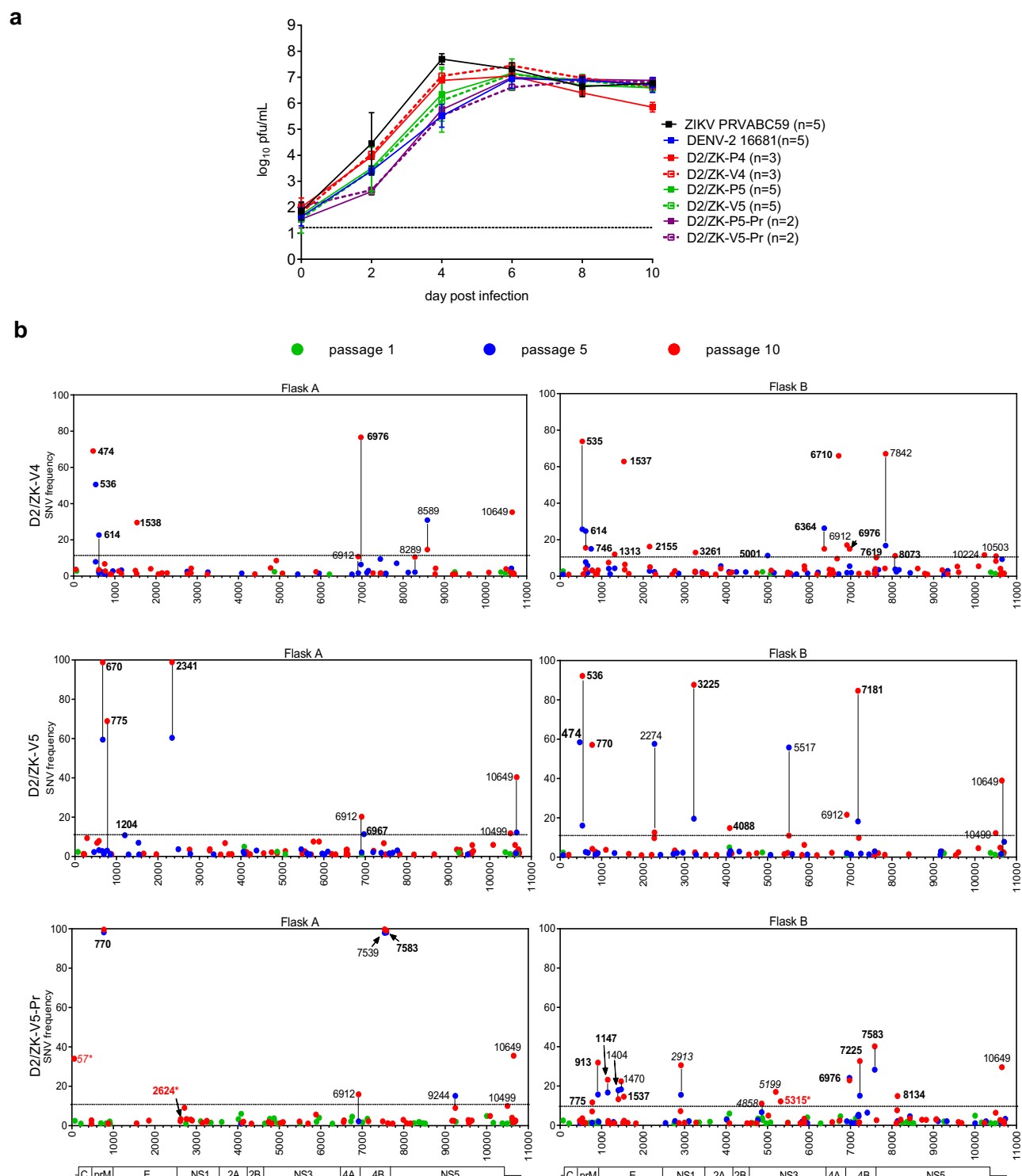

**Fig. 2 Growth kinetics and genetic stability of D2/ZK LAV candidates in Vero cells. a** Growth kinetics of D2/ZK viruses: geometric mean titers (GMT) and error bars (standard deviation, SD) of each virus group. Sample size (*n*) shown in the symbol legend indicates the number of independent culture flasks used for each virus. Dashed line represents the limit of detection (LOD) of the plaque assay. **b** Whole genome NGS of D2/ZK-V4, V5 and V5-Pr viruses at passage 1 (green), 5 (blue) and 10 (red) of 2 independent lineages (Flask A and B). SNVs with frequency ≥1% are displayed across the virus genome. SNVs at DENV-2 PDK-53 attenuation loci are highlighted in red font with an asterisk*. Nucleotide positions labeled with black font indicate SNVs ≥10% frequency (above dashed line), bold font indicates SNVs resulting in AA change. Vertical line between P5 and P10 indicates SNVs found in both passages of the lineage.

**Table 1 Single nucleotide variants (SNVs) with frequencies[a] ≥ 25% in P10 viruses.**

| SNV | | Amino acid | | D2/ZK-V4 | | D2/ZK-V5 | | D2/ZK-V5-Pr | |
|---|---|---|---|---|---|---|---|---|---|
| Position | Change[b] | Position | Change[b] | P10A | P10B | P10A | P10B | P10A | P10B |
| 57 | C→U | 5'UTR | | | | | | 34.05 | |
| 474 | U→A | prM-8 | Ser→Arg | 69.1 | | | | | |
| 535 | U→G | prM-29[c] | Leu→Val | | 73.86 | | | | |
| 536 | U→C | | Leu→Ser | | | | 92.22 | | |
| 670 | A→G | prM-40 | Thr→Ala | | | 98.82 | | | |
| 770 | C→U | M-14 | Thr→Met | | | | 57.16 | 99.56 | |
| 775 | U→C | M-16 | Ser→Pro | | | 68.94 | | | |
| 913 | A→G | M-62 | Ile→Val | | | | | | 31.89 |
| 1537 | G→A | E-195[c] | Gly→Ser | | 62.84 | | | | |
| 1538 | G→A | | Gly→Asp | 29.6 | | | | | |
| 2341 | U→A | E-463 | Phe→Ile | | | 98.93 | | | |
| 2913 | U→C | NS1-116 | | | | | | | 30.61 |
| 3225 | A→C | NS1-253 | Gln→His | | | | 87.71 | | |
| 6710 | A→G | NS4A-97 | Tyr→Cys | | 65.96 | | | | |
| 6976 | U→G | NS4B-36 | Ser→Ala | 76.62 | | | | | |
| 7181 | C→U | NS4B-104 | Pro→Leu | | | | 84.63 | | |
| 7225 | G→U | NS4B-119 | Ala→Ser | | | | | | 32.69 |
| 7539 | C→U | NS4B-223 | | | | | | 99.67 | |
| 7583 | C→U | NS4B-238 | Ser→Phe | | | | | 99.07 | 40.2 |
| 7842 | A→G | NS5-76 | | | 67.07 | | | | |
| 10649 | C→U | 3'UTR | | 35.28 | | 40.37 | 38.96 | 35.51 | 29.59 |

[a] Results from independent P10 lineages (A and B).
[b] Reference sequence → Variant of the locus. Blanks in AA change column indicate synonymous change or in UTR.
[c] Two variants located within a codon at the AA position.

reversion of 5'UTR-57 and NS1-53, respectively, whereas P10-B contained 12% reversion of NS3-250 (Fig. 2b; Supplementary Table 3). Accordingly, the D2/ZK-V5-Pr virus was deselected and excluded from most of the in vivo studies.

**Retention of DENV-2 PDK-53 vaccine virus attenuation phenotypes.** We previously defined in vitro and in vivo phenotypic markers of attenuation for the DENV-2 PDK-53 vaccine, including small plaques, temperature sensitivity, restricted growth in C6/36 cells, low transmissibility by major *Aedes* mosquito vectors, and attenuation of neurovirulence in neonatal mice[8,11,16,17,19]. All D2/ZK-V candidates were characterized for these phenotypes and compared with controls, including wt viruses, D2/ZK-P counterparts, and a clone-derived DENV-2 PDK-53 (designated as D2 PDK-53-VV45R[11,17]).

All 3 primary DENV-2 PDK-53 attenuation determinants contribute to the small-plaque phenotype of the vaccine virus when compared with its parental wt 16681 counterpart[11]. For this study, we characterized the plaque sizes of the chimeric vaccine viruses compared with their chimeric parental counterparts. All D2/ZK-V viruses exhibited significantly smaller plaques than wt ZIKV ($p < 0.0001$ for D2/ZK-V4 and -V5, $p = 0.0015$ for D2/ZK-V5-Pr), with D2/ZK-V4 producing the smallest plaques ($1.6 \pm 0.5$ mm), and D2/ZK-V5-Pr forming the largest plaques ($3.8 \pm 0.6$ mm) among the 3 candidates in Vero cell monolayers (Fig. 3a). Plaques generated by both D2/ZK-V4 and -V5 viruses were also significantly smaller than their D2/ZK-P counterparts ($p < 0.0001$). On the other hand, there was no significant difference in plaque size between D2/ZK-V5-Pr and -P5-Pr ($p = 0.1988$).

Temperature sensitivities of the viruses were examined by replication in LLC-MK2 cells at 39 °C relative to growth at 37 °C (Fig. 3b and Supplementary Fig. 1). All viruses, including wt controls were temperature sensitive to some degree. As expected, the D2 PDK-53-VV45R was more temperature sensitive than DENV-2 16681. Similarly, all D2/ZK-V viruses showed a greater degree of temperature sensitivity than their

D2/ZK-P counterparts, with the D2/ZK-V4 candidate exhibiting the greatest sensitivity.

In C6/36 cells, wt ZIKV exhibited the earliest and highest peak titer (day 4–6 post-infection, 8.7–8.9 $\log_{10}$ pfu/mL), while DENV-2 16681 reached peak titer on day 8 (8.0 $\log_{10}$ pfu/mL) (Fig. 3c). Replication of all D2/ZK-P viruses was similar to their DENV-2 16681 backbone, whereas replication of the D2/ZK-V viruses was 1000–10,000-fold lower than the D2/ZK-P counterparts, only reaching 4.2–5.1 $\log_{10}$ pfu/mL between day 8–12 post-infection. Importantly, this replication deficiency was similar to the expected replication deficiency of D2 PDK-53-VV45R[17].

We further evaluated the D2/ZK-V4 and -V5 candidates for vector transmission potential in live mosquitoes. Our previous studies demonstrated that DENV-2 PDK-53-based LAV candidates are unlikely to be transmissible by their main urban mosquito vectors, *Aedes aegypti* and *Ae. albopictus*[17,19]. We assessed infection, dissemination, and transmission capability of the D2/ZK LAV candidates in both mosquito species by measuring viral loads in mosquito bodies, legs, and saliva after oral feeding with bloodmeals containing virus.

In *Ae. Aegypti* (Poza Rica colony), 44% (18/41; geometric mean titer (GMT) = $6.0 \pm 0.4$ $\log_{10}$ pfu/body) and 25% (14/55; GMT = $4.1 \pm 1.1$ $\log_{10}$ pfu/body) of mosquitoes were infected with wt ZIKV and DENV-2, respectively (Fig. 4a, b). In contrast, 0% of mosquitoes were infected with any vaccine viruses. In *Ae. albopictus* (Lake Charles colony), 95% (36/38; GMT = $5.1 \pm 0.8$ $\log_{10}$ pfu/body) and 95% (52/55; GMT = $5.0 \pm 0.9$ $\log_{10}$ pfu/body) of mosquitoes were infected with wt ZIKV and DENV-2, respectively (Fig. 4c, d). Again, neither D2/ZK-V4 nor -V5 viruses infected any mosquitoes, while D2 PDK-53-VV45R virus infected 16% (9/55, GMT = $2.5 \pm 1.7$ $\log_{10}$ pfu/body) of mosquitoes. Because the Lake Charles *Ae. albopictus* colony was established from eggs collected in 1987[20], we conducted an additional *Ae. albopictus* study using a new EGL colony from a 2018 field collection. The results from the new colony were similar to the Lake Charles colony (Fig. 4e, f).

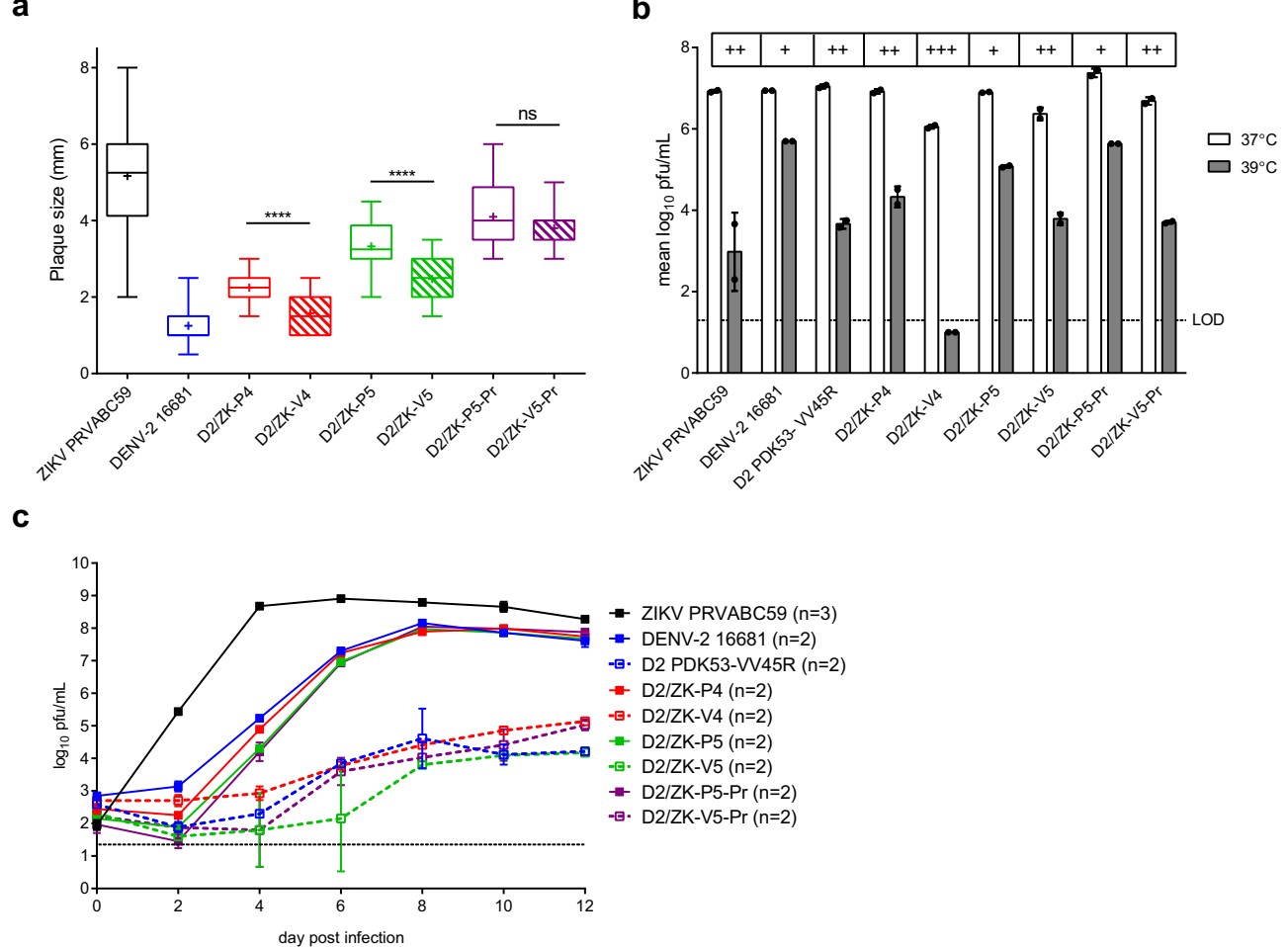

**Fig. 3 In vitro attenuation markers of vaccine candidates. a** Virus plaque diameter in Vero cells. Box and whiskers plot shows the range of plaque sizes observed for each virus ($n = 12$ independent plaques for wt ZIKV; $n = 20$ plaques for other viruses). The whiskers represent the maximum and minimum diameters, the box is bound by the 25th and 75th percentiles with the median indicated by a solid line and the mean indicated with a cross. A two-tailed, unpaired $t$ test was used to determine statistical differences between P and V viruses. ****$p < 0.0001$, ns not significant ($p = 0.1988$). **b** Temperature sensitivity of viruses in LLC-MK2 cells. Peak geometric mean titers (GMTs) with standard deviation (SD) from a growth kinetics study (Supplementary Fig. 1) are shown for each virus. The $\log_{10}$ differences of peak GMTs between cultures at 37 °C and 39 °C are indicated as +(1–2.5 $\log_{10}$), ++(2.5–4 $\log_{10}$) or +++(>4 $\log_{10}$). Dashed line represents the limit of detection (LOD) of the plaque assay. Black dots show data points from 2 independent cultures ($n = 2$). **c** Growth kinetics of viruses in C6/36 cells: GMTs and SD of each virus group over the time course of the experiment. Number ($n$) of independent cultures conducted is indicated in the symbol legend for each virus. Dashed line represents the LOD of the plaque assay.

Because virus dissemination to legs is only detectable in mosquitoes with positive body infection, only infected mosquitoes were analyzed for dissemination. Similarly, only saliva samples collected from mosquitoes with positive dissemination were analyzed for potential transmission. Positive dissemination and transmission were observed for wt controls in both *Aedes* mosquitoes (Fig. 4), confirming these mosquito colonies are competent for both wt ZIKV and DENV-2 as expected.

To evaluate mouse neurovirulence of the vaccine candidates, we first established a highly sensitive newborn mouse neurovirulence model for wt DENV-2 and ZIKV using commercially available, timed-pregnant CD-1 mice. We determined that 0–2 day old newborns suffered 100% morbidity following intracranial (i.c.) challenge with $10^4$ pfu of wt ZIKV or DENV-2 16681, while 70–100% of 4–5 day old mice were resistant to this i.c. challenge dose (Supplementary Fig 2). To investigate the neurovirulence of the vaccine candidates, 1-day-old CD-1 mice were i.c inoculated with $10^4$ pfu of virus. As expected, 0% survival was observed among mice inoculated with either wt ZIKV (average survival time (AST) ± SD = 9.8 ± 0.4 days) or DENV-2 (10.8 ± 0.4 days), whereas

all mice inoculated with D2 PDK-53-VV45R vaccine or chimeric D2/ZK viruses survived (Fig. 5a). Mice challenged with wt viruses showed minimal average weight gain ~0–2 days before reaching the morbidity score for termination (Fig. 5b and inset). Importantly, continued weight gain without sign of illness was observed in animals inoculated with PBS or any of the chimeric D2/ZK viruses, including all D2/ZK-V viruses and their counterpart D2/ZK-P viruses (Fig. 5).

**Safety, immunogenicity, and protective efficacy in adult AG129 mice**. The IFNα/β and γ receptor knockout AG129 mouse is highly susceptible to ZIKV infection and recapitulates many important disease features observed in humans, including neurological disease, brain pathology, and virus dissemination to the male and female reproductive tract[21–23]. In addition, AG129 mice have previously been used for pre-clinical evaluation of TDV[9,17]. We first compared the immunogenicity and protective efficacy after one and two immunizations using the D2/ZK-V5 candidate vaccine. Intraperitoneal immunization with $10^4$ pfu of D2/ZK-V5 and D2/ZK-P5 viruses resulted in similar and robust neutralizing

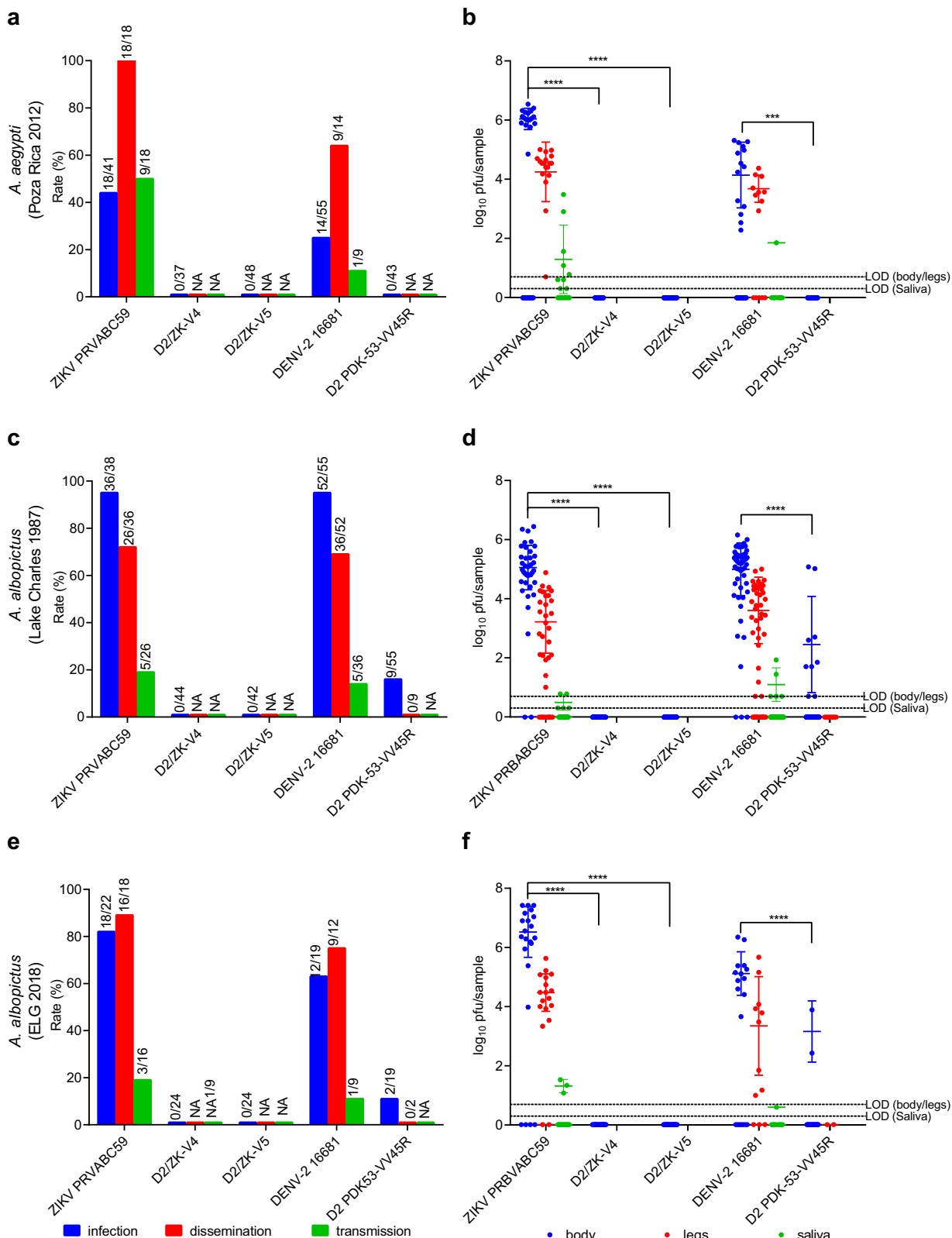

antibody (NAb) titers 40 days post-primary immunization (Fig. 6). A second immunization did not result in significantly higher NAb titers on day 68 compared with titers of the groups receiving only one immunization (Fig. 6a). After wt ZIKV challenge, the naïve control animals had high viremia 3 days post-challenge (GMT = 6.5 $log_{10}$ pfu/mL), significant weight loss, and 100% of the mice succumbed to infection (AST ± SC = 16.0 ± 4.4 days) (Fig. 6b–d).

In contrast, all D2/ZK-V5 immunized animals (both single- and double-immunization groups) were fully protected without any detectable viremia, weight loss, or illness.

We next evaluated safety and compared the immunogenicity and efficacy of a single immunization (at $10^4$ and $10^3$ pfu doses) of all D2/ZK-V vaccine candidates (Fig. 7). Unlike wt ZIKV that is lethal for adult AG129 mice at $10^4$ pfu, all D2/ZK-V viruses and

**Fig. 4 Mosquito infection, dissemination, and transmissibility potential of vaccine candidate viruses. a**, **b** Ae. aegypti (Poza Rica 2012), **c**, **d** Ae. albopictus (Lake Charles 1987), **e**, **f** Ae. albopictus (ELG 2018). **a**, **c**, **e** Infection, dissemination, and transmission rates. The number of positive bodies out of the total number of mosquitoes is indicated above the blue bar (infection). The number of mosquitoes with positive legs among the mosquitoes with virus-positive bodies is indicated above the red bar (dissemination). The number of mosquitoes with positive saliva among mosquitoes with positive legs is indicated above the green bar (transmission). NA indicates not analyzed, as no sample was tested due to previous negative infection or dissemination of the virus. A 0% rate was drawn at 1% for illustration purposes. **b**, **d**, **f** Virus titers in mosquito bodies (blue), legs (red) and saliva (green). GMT and SD are based on positive samples for each group. The dashed lines represent the LOD for each sample type. All tested samples, including negative ones (assigned as and graphed at 0) were included for statistical comparison between wt and vaccine groups. A two-tailed, unpaired $t$ test was used to determine statistical differences. ***$p = 0.0005$, ****$p < 0.0001$. The sample size for each group used to derive statistics is indicated by the denominator above the bars in **a**, **c** and **e**.

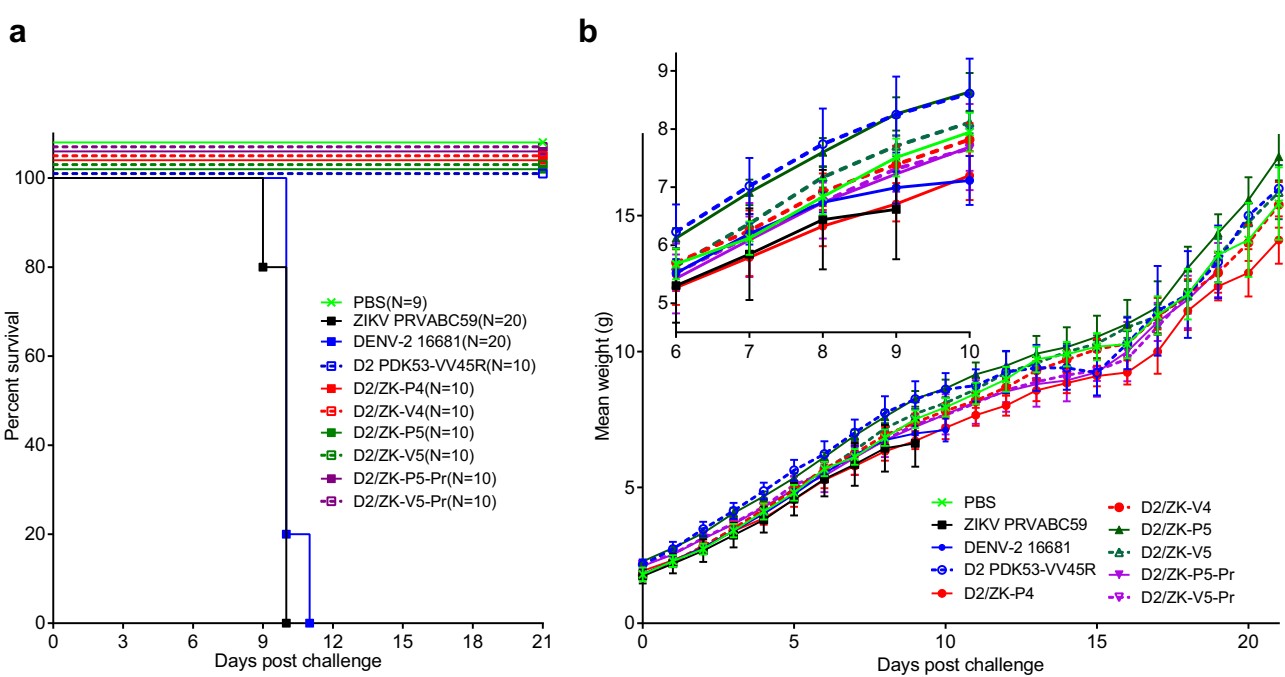

**Fig. 5 Chimeric viruses are non-neurovirulent in neonatal CD-1 mice. a** Kaplan–Meyer survival curves of 1-day-old CD-1 mice inoculated i.c. with $10^4$ pfu of virus. **b** Group mean weight ±SD post virus challenge. Inset figure shows a close-up of the graph between 6 and 10 days post challenge. The number ($n$) of mice in each virus group is indicated in the symbol legend of figure **a**.

their P counterparts tested at $10^4$ pfu were non-virulent for AG129 mice at this dose. Each of the D2/ZK-V viruses elicited a similar and robust NAb response, with GMTs ranging 3.0–3.9 $\log_{10}$ by day 70 post-immunization (Fig. 7a). GMTs were not significantly different between the $10^3$ and $10^4$ pfu doses of each virus, nor between measurements at 40- and 70-days post-immunization among vaccine candidates. NAb titers elicited by the D2/ZK-V viruses were similar to the titers of their respective D2/ZK-P viruses, suggesting the chimeric viruses based on the DENV-2 PDK-53 backbone were as immunogenic as those containing a wt DENV-2 backbone in mice. Unlike PBS control animals (AST ± SD = 14.1 ± 4.0 days), all groups of D2/ZK immunized animals were protected from lethal wt ZIKV challenge as indicated by absence of viremia, weight loss or illness (Fig. 7b–d). These results demonstrated that a single $10^3$ pfu-dose immunization with any of the D2/ZK-V candidates was sufficient to protect mice from lethal ZIKV challenge.

**Safety, immunogenicity, and efficacy of D2/ZK LAV candidates in nonhuman primates.** Non-human primates (NHPs) exhibit many characteristics of ZIKV infection in humans, including rapid onset of viremia that clears as NAbs develop, presence of ZIKV RNA in saliva, urine, reproductive tissues, cerebrospinal fluid and brain, as well as prolonged viremia during pregnancy, and pregnancy complications including miscarriage and congenital defects, making NHP a highly relevant animal model for ZIKV vaccine evaluation[24–28]. Flavivirus-seronegative Indian rhesus macaques were immunized subcutaneously (s.c.) with $10^4$ pfu of D2/ZK-V4 (one immunization on day 0) or D2/ZK-V5 (two immunizations on days 0 and 91). Attenuation of the vaccine candidates was assessed by measuring viral RNA (vRNA) in serum (vRNAmia) post-vaccination. Following initial vaccination, only 1 of 6 animals immunized with D2/ZK-V4 (animal-K940) or -V5 (animal-K289) were positive for vRNAmia at a low level (3.3–3.5 $\log_{10}$ copies/mL) for 1 day (Fig. 8a). None of the animals in the D2/ZK-V5 group exhibited vaccine vRNAmia following the second immunization. As unvaccinated animals developed vRNAmia for 3–4 days following ZIKV challenge with peak GMT at a level roughly 10-fold higher (day 4, 4.7 $\log_{10}$ copies/mL) (Fig. 8c top panel), these data demonstrated that both LAV candidates were attenuated for replication in rhesus macaques.

Immunogenicity of the LAV candidates was evaluated by measuring NAb against ZIKV using a reporter virus-based micro-focus reduction neutralization test (R-mFRNT) (Fig. 8b). Seroconversion was detected as early as day 9 in D2/ZK-V4 immunized animals, and 5/6 animals seroconverted by day 14 post-immunization (Fig. 8b, middle panel). The NAb titers

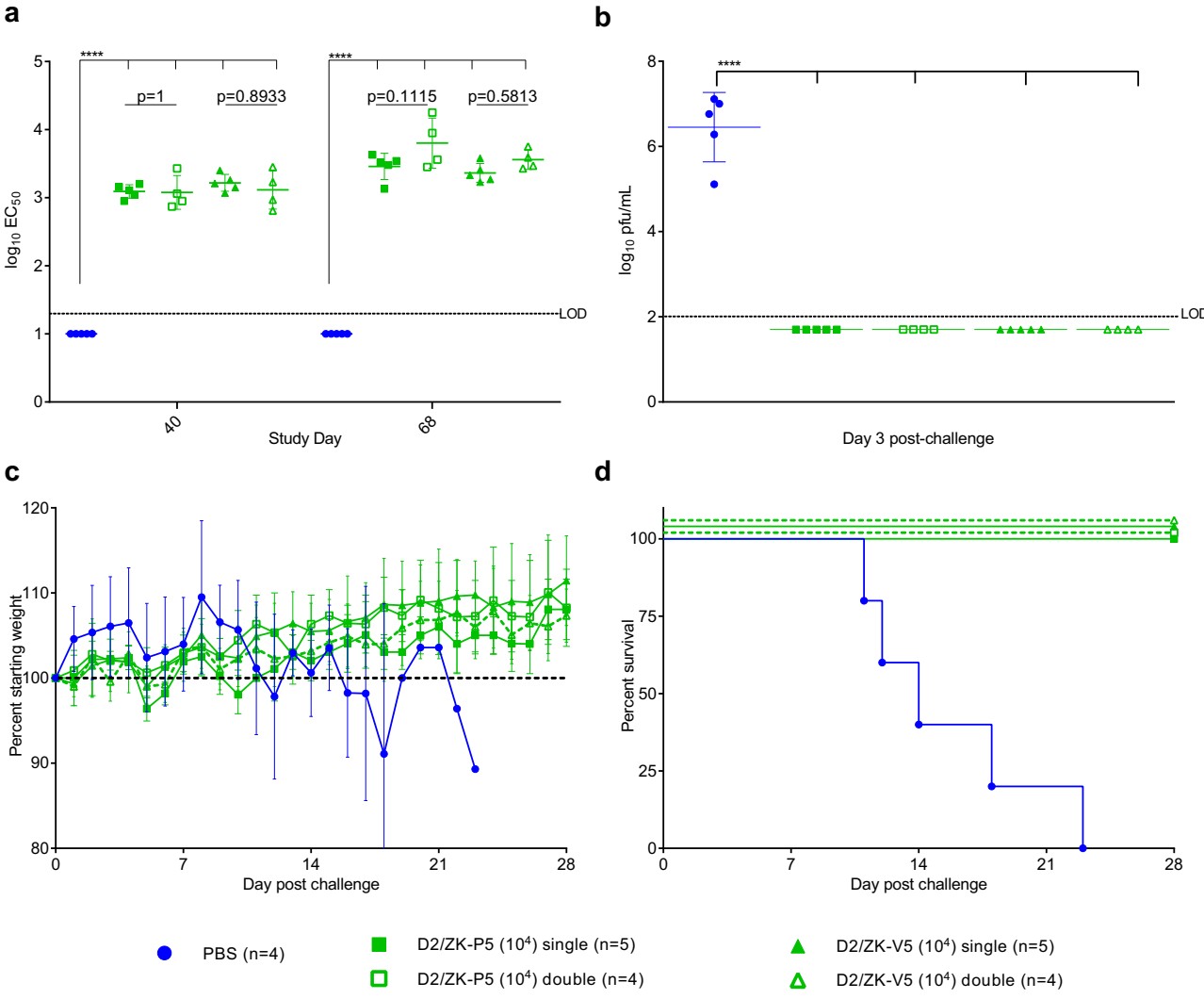

**Fig. 6 Protective efficacy and immunogenicity of single- or double-dose of D2/ZK-P5 and -V5 virus in AG129 mice.** Separated groups of mice were immunized once (day 0) or twice (day 0 and 42), and challenged with $10^4$ pfu of wt ZIKV on day 70. **a** NAb titers of immunized mice at 40 and 68 days after primary immunization. **b** Viremia after wt ZIKV challenge. **c** Weight changes after ZIKV challenge. **d** Kaplan–Meyer survival curves after lethal ZIKV challenge. Horizontal bars (in **a** and **b**) represent the group GMT, error bars represent SD. Negative samples are plotted at the half value of the assay LOD and have been included in determination of GMTs and statistical analysis (one-way ANOVA with Tukey's test to correct multiple comparisons). ****$p < 0.0001$, Not significant: $p > 0.05$ (exact $p$ values provided in graph). Number ($n$) of mice used in each group is indicated in the symbol legend.

peaked between days 14–45, and the GMT on day 30 was $2.4 \pm 0.3 \log_{10}$ EC$_{50}$. The NAb titers were generally maintained through day 115, followed by a gradual decline to day 186, with the exception of animal-K653 whose titer began waning at day 30 and became undetectable by day 91. In the D2/ZK-V5 group, 2/6 and 5/6 immunized animals seroconverted by 11 and 14 days post-immunization, respectively, with titers peaking between days 14–45 and group GMT at $2.2 \pm 0.7 \log_{10}$ on day 30 (Fig. 8b, lower panel). Half of the animals had an anamnestic response on day 115, 24 days post-second immunization, and NAb titers gradually declined in all animals by day 157 before plateauing through day 186. One animal, DGE2, did not seroconvert following primary D2/ZK-V5 immunization but seroconverted after the second immunization. In addition to the R-mFRNT, a reporter virus particle (RVP) microneutralization assay was also employed (Supplementary Fig. 2). Similar kinetics of the NAb response were observed with both assays. Overall, both D2/ZK-V4 (one immunization) and -V5 (two immunizations) LAVs elicited a rapid NAb response that persisted for at least 6 months in most animals.

To assess protective efficacy, vaccinated animals were challenged with $10^4$ pfu of wt ZIKV at 6 months post-primary immunization (study day 186). All animals in the PBS group developed vRNAmia for 3–4 days (peak GMT = $4.7 \log_{10}$ copies/mL, on day 4) (Fig. 8c). In contrast, none of the vaccinated animals developed detectable ZIKV vRNAmia post-challenge with the exception of animal-K653 in the D2/ZK-V4 group, which had vRNAmia only on day 2 post-challenge (3.7 log copies/mL; Fig. 8c). This vRNAmia level was substantially lower and of shorter duration compared with the PBS control group. Of note, NAbs in animal-K653 declined to an undetectable level at the time of challenge. A substantial and rapid anamnestic NAb response was observed in both D2/ZK-V immunized groups following wt ZIKV challenge (Fig. 8b). Overall, D2/ZK-V immunization resulted in a rapid onset of NAb titers that lasted for at least 6 months and all animals with detectable NAbs were protected against vRNAmia following challenge with wt ZIKV.

## Discussion

We constructed and characterized chimeric ZIKV LAV candidates. To develop these chimeric D2/ZK LAV candidates, we first

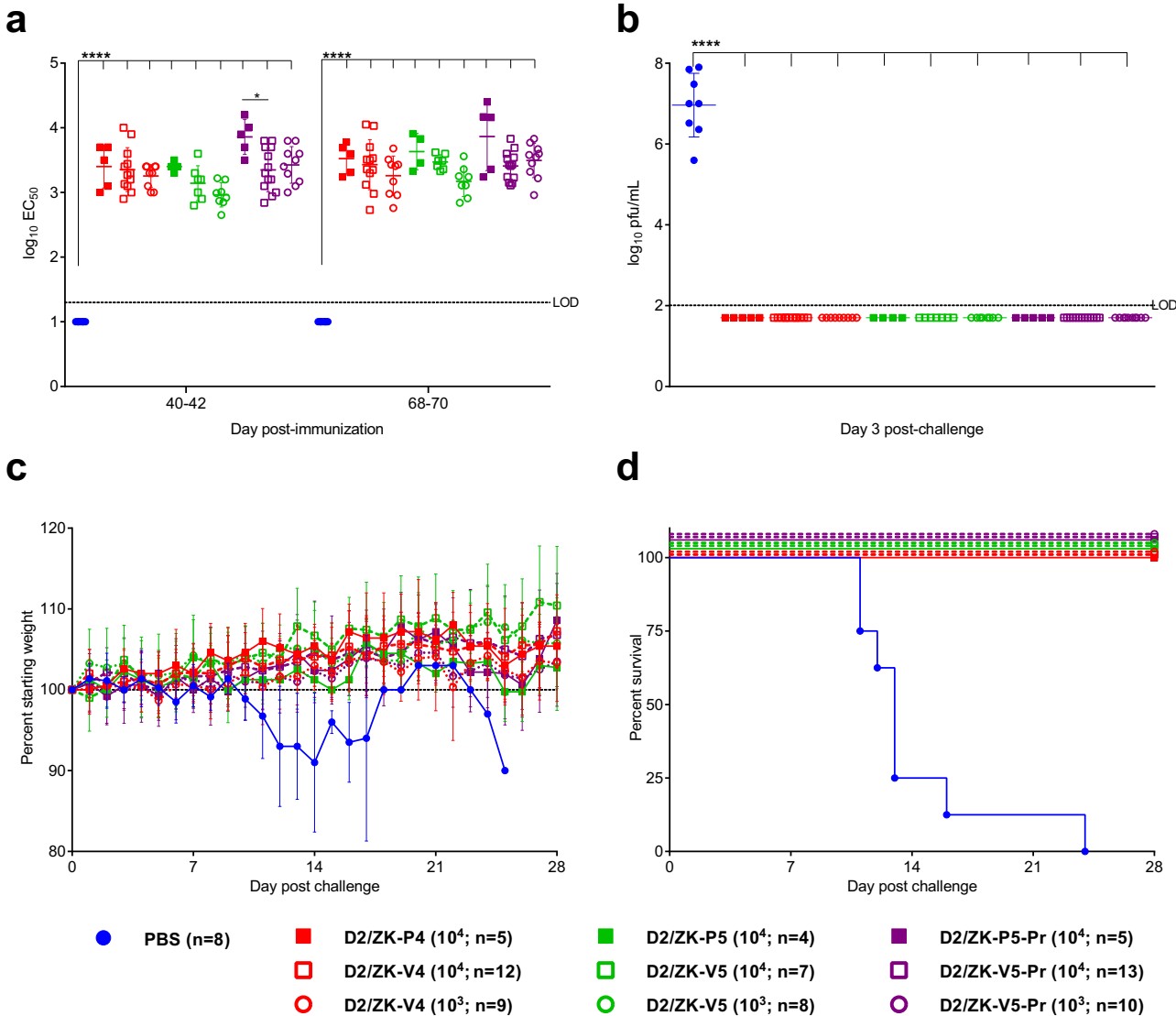

**Fig. 7 Single dose of D2/ZK LAV candidates protect AG129 mice from lethal ZIKV challenge.** D2/ZK-V candidates were tested at 2 single-dose levels, $10^3$ and $10^4$ pfu. Their parental chimeric control viruses, D2/ZK-P, were tested at $10^4$ pfu dose only. The naïve group received PBS. All mice were challenged with $10^4$ pfu of wt ZIKV on day 70. **a** NAb titers of mice at 40–42 days and 68–70 days after the single immunization. **b** Viremia after wt ZIKV challenge. All immunized animals were protected from viremia. **c** Weight changes after ZIKV challenge. **d** Kaplan–Meyer survival curves after wt ZIKV challenge. Horizontal bars (in **a** and **b**) represent the group GMT, error bars represent standard deviation. Negative samples are plotted at the half value of the assay LOD and have been included in determination of GMTs and statistical comparisons (one-way ANOVA with Tukey's test to correct multiple comparisons). *$p = 0.186$, ****$p < 0.0001$. Number ($n$) of mice used in each group is indicated in the symbol legend. With exception of the D2/ZKV-V5-Pr group showing significantly lower NAb titers than its P counterpart group on day 40–42, no significant differences were observed between groups that received D2/ZK-V and -P viruses.

optimized the chimeric constructs for growth and genetic stability in Vero cells. We additionally demonstrated that the vaccine candidates retain all DENV-2 PDK-53 vaccine attenuation phenotypic markers, including attenuation for replication, dissemination, and transmission potential in the 2 major mosquito vectors of ZIKV and DENV. Finally, the results demonstrate that the vaccine candidates are immunogenic and protect mice and NHPs against ZIKV challenge.

Tissue culture adaptation for recovery of viable chimeric flaviviruses has been reported in the development of ZIKV LAV candidates utilizing the YFV-17D backbone[29,30]. Additionally, our experience with PDK-53-based chimeric D2/WNV and some of the chimeric DENV constructs also showed that cell adaptation was required to optimize chimeric virus yield and genetic stability in Vero cells[8,9]. In this study, we identified 5 potential Vero cell

adaptive mutations, 3 of which are near the chimeric junction in the E transmembrane domain (E-TMD) preceding the NS1 gene. The flavivirus E-TMD contains 2 membrane anchor elements, TM1 and TM2[31], and the junction site of D2/ZK is located within the TM2. Therefore, the chimeric viruses contain TM1 of ZIKV, and a chimeric TM2 composed of 9 AA from ZIKV followed by 13 AA from DENV-2. Adaptive mutations (E-Q465R in ZIKV TM1, E-I484T in ZIKV TM2, and E-I493F in DENV-2 TM2) likely compensated for defective interaction between TM1 and TM2 caused by chimerization of the 2 heterologous viruses. The other 2 adaptive mutations (NS2A-K99N and NS4A-D23N) occurred in the NS proteins that are critical for RNA replication and virion assembly[32]. None of these 5 Vero-adaptive substitutions were identical to those identified in other chimeric LAVs based on the same platform, and an attempt to incorporate a

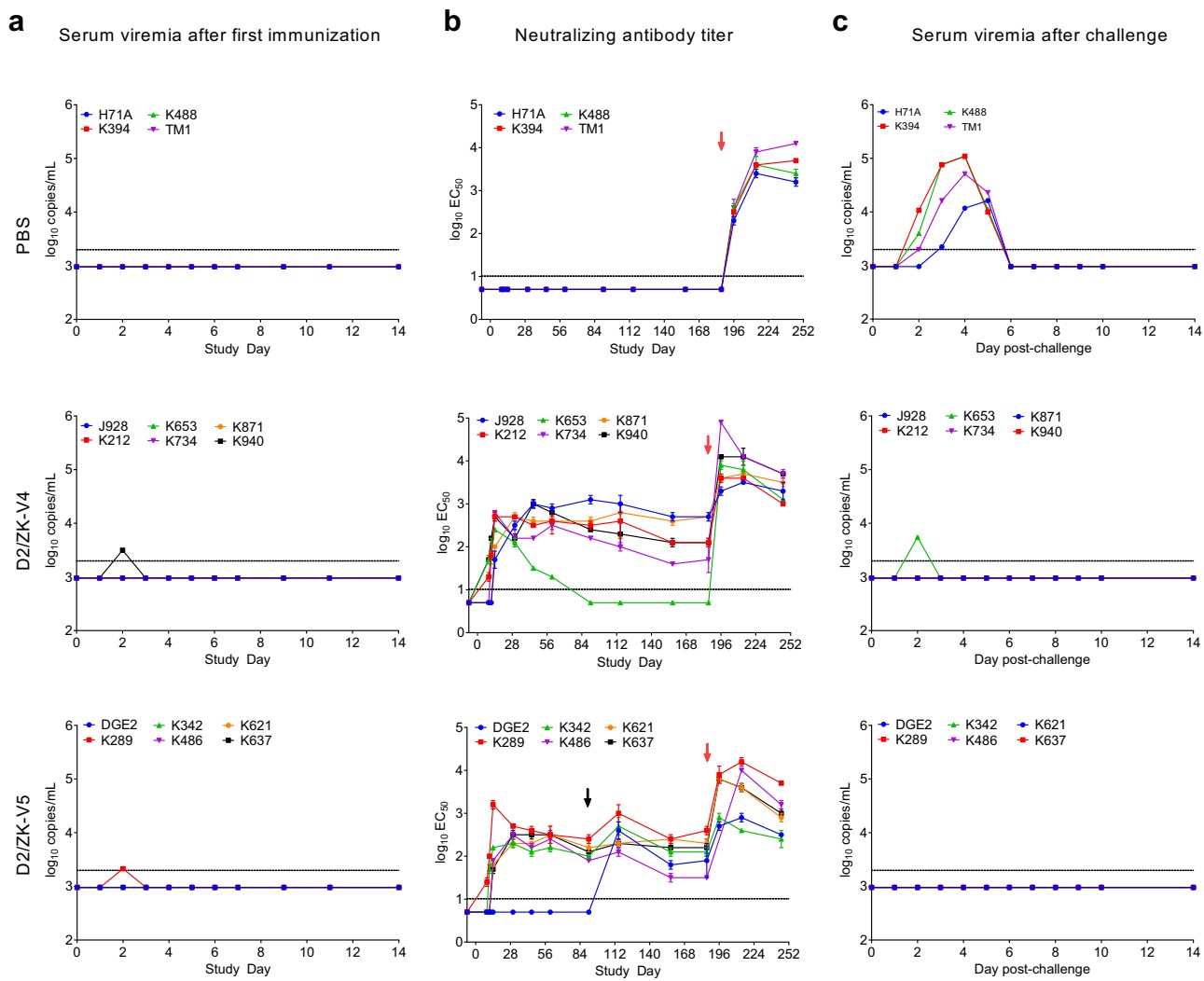

**Fig. 8 D2/ZK-V4 and D2/ZK-V5 vaccination protects rhesus macaques following ZIKV challenge.** Animals were immunized on day 0 (D2/ZK-V4 and -V5) and day 91 (D2/ZK-V5 group only) before challenge with $10^4$ pfu of wt ZIKV on day 186. **a** Vaccine vRNAmia after immunization. **b** NAb kinetics (by R-mFRNT) after vaccination and wt ZIKV challenge. Black arrow indicates day of boost (D2/ZK-V5 group only) and red arrow indicates day of challenge. **c** ZIKV vRNAmia after wt ZIKV challenge. Negative samples were plotted at the half value of the assay LOQ or LOD. Dashed lines indicate assay limit of quantitation (LOQ, 3.3 $\log_{10}$ copies/mL; in **a** and **c**) or LOD (1.0 $\log_{10}$ $EC_{50}$, in **b**). Graphs in **a** and **c** show the mean of qRT-PCR results from samples tested in triplicate. Graph in **b** represents mean and SD of 2 ($n = 2$) independent titration experiments (samples were titrated in triplicate in each experiment).

previously identified Vero-adaptive NS2A-22 mutation in D2/WNV[8] failed to improve the viability of D2/ZK. These results indicate that optimization of each chimeric flavivirus construct requires empirical investigation, even if based on the same attenuated flavivirus platform.

The D2/ZK-V4 and -V5 LAV candidates exhibited a smaller plaque phenotype in Vero cells, greater temperature sensitivity in LLC-MK$_2$ cells, and reduced replication in C6/36 cells relative to their respective D2/ZK-P viruses. Lack of transmissibility by mosquito vectors is an important environmental safety criterion for mosquito-borne flavivirus LAVs. To be transmissible by mosquito vectors, virus ingested by the mosquito must overcome the mosquito midgut barriers, including the midgut infection barrier (establishing productive infection) and the midgut escape barrier (permitting dissemination to other tissues), and be expectorated in mosquito saliva. Although flavivirus replication in C6/36 cells does not always predict mosquito competency in vivo, a positive correlation of the DENV-2 PDK-53 vaccine attenuation determinants between C6/36 cells and whole mosquitoes has been well established[11,33], and the decreased

replication efficiency of chimeric DENV-2 PDK-53-based LAVs (usually at least 1000-fold lower) in C6/36 cell culture relative to their chimeric parental viruses has reliably predicted their attenuation in whole mosquitoes[17,19]. In this study, the attenuation of D2/ZK-V4 and -V5 in C6/36 cells also correlated with the outcomes in 3 tested mosquito colonies. While these mosquitoes were competent for transmission of wt ZIKV and DENV-2, both vaccine candidates failed to establish a productive midgut infection that is a prerequisite for virus dissemination or transmission. These results are consistent with our previous findings showing chimeric flavivirus LAVs based on the DENV-2 PDK-53 platform have very low potential for mosquito transmission[17,19].

The suckling mouse model utilizing i.c. virus challenge has been widely used to evaluate neurovirulence of flaviviruses and safety of their LAV candidates[8,9,11,16,17,34,35]. High concordance between the neurovirulence profiles in suckling mice (mortality rate) and rhesus monkeys (histopathological brain lesion scores) has been demonstrated for flaviviruses and their LAV viruses, suggesting the newborn mouse model could substitute as an appropriate safety test for neurovirulence of flavivirus vaccines[35].

All chimeric D2/ZK-P and D2/ZK-V viruses were found to be highly attenuated in neonatal CD-1 mice by i.c. challenge and adult AG129 mice by i.p. challenge, indicating virus chimerization itself plays a role in mouse attenuation, as previously demonstrated[9,16]. The D2/ZK-V5-Pr virus with an additional prM-N17S substitution was included for its reported association with attenuation in mouse neurovirulence[18]. Paradoxically, the introduction of prM-N17S resulted in a chimeric virus that lost the smaller plaque phenotype in comparison with D2/ZK-V5 or its D2/ZK-P5-Pr counterpart and was less genetically stable than the other 2 candidates. Since both D2/ZK-V4 and D2/ZK-V5 were non-virulent in suckling and adult mice, the addition of the prM-17S mouse attenuation locus in the context of the D2/ZK-V5 was deemed unnecessary.

A chimeric ZIKV LAV based on a similar DENV-2 PDK-53 backbone (VacDZ) was recently described[36]. Differences between our D2/ZK-V candidates and VacDZ include the number of DENV-2 PDK-53 specific mutations, the C-terminal chimeric junction, and the inclusion of Vero-adaptive substitutions. Although we have determined the 3 major PDK-53 attenuation loci in vitro and in animal models, we chose to include other PDK-53 specific loci as we cannot rule out their potential contributions to attenuation or virus stability in humans. Inclusion of Vero cell adaptive mutations in the D2/ZK candidates yielded about 10-fold higher virus titers in Vero cells ($\geq$ 7 log pfu/mL), compared with ~6 $\log_{10}$ pfu/mL for VacDZ. Despite these differences, the two studies support the genetic stability of the 3 major DENV-2 PDK-53 attenuation loci, the retention of attenuation phenotypes, and the safety and efficacy of chimeric DENV-2/ZIKV vaccine candidates in mouse models; and we extended these findings by demonstrating genetic stability in a Vero cell line used for vaccine manufacturing, attenuation in live mosquitoes, and protective efficacy in non-human primates.

While our analysis of immunogenicity was limited to NAbs, active and passive protection studies in mice and NHP models of ZIKV infection support NAbs as an immune correlate of protection[5,6,37–43]. However, other immune responses, such as CD8+ cells may also contribute to protective immunity conferred by the chimeric D2/ZK LAV candidates. The primary targets of CD8+ responses against ZIKV and DENV are the NS proteins[44] and the cross-reactive CD8+ T cells induced by DENV have been shown to be protective against ZIKV[45–47]. In addition, TDV (TAK-003) has been shown to elicit potent and durable CD8+ T cell immunity predominantly targeting NS1, NS3, and NS5 with significant cross-reactivity against all 4 DENV serotypes[48]. Therefore, cross-reactive CD8+ responses elicited by the DENV-2 NS proteins of the chimeric D2/ZK LAV may also contribute to protective immunity against ZIKV. However, further studies are required to evaluate such cross-protection potential, as one other study indicated that T-cell immunity induced by DENV had limited cross-reactivity to other flaviviruses[49].

There are concerns that pre-existing flavivirus immunity could lead to antibody-dependent enhancement (ADE) of subsequent flavivirus infection and increased disease severity. Thus far, human cohort and epidemiological studies have indicated prior DENV immunity does not enhance ZIKV infection or illness severity, but immunity from a prior DENV and/or ZIKV infection can enhance future severe dengue disease for some DENV serotypes[50,51]. As the chimeric D2/ZK LAV is based on a DENV-2 backbone, the CD8+ T-cell and NS1 antibody responses elicited by the DENV-2 NS proteins have been shown to provide cross-protection against other DENVs[48,52]. Such cross-protection may mitigate the risk of illness enhancement following a DENV infection. In addition, ZIKV vaccine administered with or after a tetravalent DENV vaccine could be a strategy to avoid the potential risk of ADE of subsequent DENV infection. Because the TDV (TAK-003) vaccine is based on the same PDK-53 platform as the D2/ZK LAV candidate, these vaccine viruses may be compatible for combination as a pentavalent vaccine, or for simultaneous or sequential vaccination. Ultimately, discerning the potential immune interactions between DENV and ZIKV vaccines and identifying ADE mitigation strategies will require more thorough investigation during clinical development.

Utilizing the well-established chimeric DENV-2 PDK53 vaccine platform, we have successfully generated ZIKV LAV candidates that are genetically stable and are immunogenic and efficacious in mouse and NHP models of ZIKV infection. While we determined the D2/ZK-V4 and -V5 candidates were reasonably stable after 10 Vero cell amplifications, the potential phenotypic consequences of SNVs that evolved after cell passages were not evaluated in this study. Since random SNVs throughout the viral genome are common during cell culture amplification of any RNA virus, careful genetic and phenotypic characterization of the clinical D2/ZK vaccine seeds produced during the cGMP-manufacturing process (from the initial rederivation of virus from the cDNA clone to multiple purification and amplification steps in large scale cell culture) would be necessary to ensure any SNVs that evolved in clinical seeds would not alter the pre-defined safety and immunogenicity of the vaccine virus prior to human clinical trials. Such evaluation of commercially produced vaccine seeds is exemplified by the comprehensive characterization we previously conducted for the cGMP-manufactured TDV clinical seeds[17]. Finally, as a primary objective for a ZIKV vaccine would be to reduce the risk of congenital infection, we are currently conducting additional animal studies evaluating fetal protection from vertical and sexual transmission of ZIKV.

## Methods

**Ethics statement**. All animal studies were performed in accordance with the Public Health Service Policy on Humane Care and Use of Laboratory Animals, and the Guide for the Care and Use of Laboratory Animals. The mouse study and protocols were approved by the Institutional Animal Care and Use Committee (IACUC) at the Division of Vector-Borne Diseases, CDC (DVBD/CDC protocol #16-017). The study in non-human primates was reviewed and approved by the IACUCs at Alpha Genesis, Inc (protocol #19-04) and DVBD/CDC (protocol #19-009).

**Cells and viruses**. DENV-2 16681 (GenBank: U87411), DENV-2 PDK-53 (Gen-Bank: U87412.1)[10] and ZIKV Brazil SPH 2015 (GenBank: KU321639) served as the genotypes for construction of the chimeric D2/ZK viruses. ZIKV PRVABC59 (GenBank: KU501215.1) was used as wt ZIKV control in the study. Both PRVABC59 and SPH2015 are contemporary ZIKV strains of Asian lineage and they differ by only a single AA (E-23I in SPH2015 and E-23V in PRVABC59) within prM-E. The DENV-2 16681 virus and D2 PDK-53-VV45R virus (clone-derived DENV-2 PDK-53-V variant[17], equivalent to TDV-2 of TAK-003) were included as DENV-2 parental and vaccine controls, respectively. All viruses were amplified under 5% $CO_2$ at 37 °C in Vero or LLC-MK2 cells using DMEM medium or at 28 °C in C6/36 *Ae. albopictus* cells using YE-LAH medium (0.033% yeast extract, Earle's balanced salt solution, 0.165% lactalbumin hydrolysate, 25 mg of gentamicin sulfate and 1.0 mg of amphotericin B per liter). The WHO Vero reference cell bank (RCB) 10-87, which has been subjected to a broad range of tests to establish its suitability for vaccine production[15], was used for virus culture, growth kinetics, genetic stability studies, and R-mFRNT. A Vero cell line maintained by the CDC cell culture laboratory was used for virus plaque assays.

**Construction of D2/ZK chimeric cDNA plasmids**. The chimeric D2/ZK cDNA construct design was identical to previously described D2/WN-V1 with prM-E of WNV replaced by that of ZIKV, and the full-length chimeric cDNA was generated by ligation of two intermediate clones[8]. Initially, a synthesized cDNA cassette containing the prM-E of ZIKV Brazil SPH 2015 (flanked by SacII and NgoMIV) was purchased from GeneArt (Sigma Aldrich) and cloned into pD2i/WN-P-SA and pD2i/WN-E-SA clones[8], resulting in pD2i/ZK-P and pD2i/ZK-V clones, respectively. The 5'-end clones were ligated with the 3'-end pD2i-P or -V clones[9] to form full-length chimeric cDNA of D2/ZK-P0 and D2/ZK-V0, and then subsequently digested with XbaI to generate the 3'-terminal end needed for in vitro transcription of vRNA using an Ampliscribe T7 High Yield Transcription kit (Epicenter) with 5'-A RNA cap structure analog (New England Biolab)[16]. Additional clones with various combinations of possible Vero-adaptive mutations were constructed by

introducing the substitutions into these initial cDNA clones using a Q5 site-directed mutagenesis kit (New England Biolabs).

**Recovery of D2/ZK chimeric viruses**. In vitro transcribed vRNA was electroporated into C6/36 and Vero cells using a BioRad GenePulser XCell as described previously[53]. Transfected cells were cultured as described above, and an immunofluorescence assay (IFA) with 1:200 dilution of anti-flavivirus mouse monoclonal antibody 4G2 (Arbovirus Reference Collection/CDC), and 1:15,000 dilution of FITC-conjugated goat AffiniPure anti-mouse IgG H + L (Jackson ImmunoResearch Laboratories, Inc.) was used to confirm expression of viral E proteins. Culture supernatant was analyzed by qRT-PCR using a Titan One Tube RT-PCR kit (Sigma Aldrich) targeting a segment containing both DENV-2 and ZIKV genome regions. An increasing quantity of IFA-positive cells and/or chimeric genome copy numbers during cell culture indicated successful generation of infectious chimeric virus. All chimeric viruses recovered from transfected cells were further amplified once in Vero or C6/36 cells, and then fully sequenced to confirm the working virus stocks were identical to their respective cDNA clones.

**Virus plaque assay**. Titrations were performed as described[5] with some modifications. Briefly, serial dilutions of virus were inoculated into 6-well plates (100 μL/well) containing just-confluent Vero cell monolayers and adsorbed at 37 °C for 1–1.5 h. After adding agarose overlay medium, the plates were incubated for 4–7 days at 37 °C/5% CO$_2$ before adding a second agarose overlay medium containing neutral red. Following addition of the second overlay, plaques were counted for 2–3 days. Plaque size measurement for comparison was performed similarly with the second overlay medium added on day 5 and size measurement (12–20 plaques/virus) on day 7 post-infection for all viruses performed in the same experiment.

**Plaque isolation and Sanger sequencing for identification of Vero-adaptive mutations**. Multiple, well-isolated viral plaques under agarose overlay medium were picked and amplified in a 25-cm$^2$ flask of Vero cells to generate stocks for Sanger sequencing to identify potential Vero-adaptive mutations. Identified mutations were incorporated into D2/ZK-V, and the mutant viruses recovered from transfected Vero cells were evaluated for plaque phenotype and genome sequencing to identify those associated with uniform plaque phenotype and stable genome sequences. This process was repeated until a desirable chimeric virus was achieved. Sanger sequencing of the viral genome was performed with overlapping RT-PCR amplicons (Roche Titan One-Tube RT-PCR) generated from extracted vRNA (QIAamp viral RNA extraction kit). Both strands of cDNA amplicons were sequenced using the ABI Prism BigDye v3.1 Terminator Cycle Sequencing kit and virus-specific primers (Supplementary Table 4) (Integrated DNA Technologies and Biotechnology Core Facility/CDC) on a thermocycler. The sequencing reactions were purified with a BigDye XTerminator Purification Kit and analyzed by an ABI 3130XL Genetic Analyzer. Sequence contigs were assembled and analyzed using DNAStar Lasergene software (v15).

**Genetic stability in Vero cells and next-generation sequencing**. Chimeric candidate vaccine viruses were serially amplified in Vero cells with serum-free DMEM for 10 passages at a low multiplicity of infection (MOI) to assess their genetic stability. Vero cells were inoculated with D2/ZK-V4, V5 and V5-Pr viruses at MOI of 0.001 for the first passage (P1). For the following 9 passages, duplicate 75-cm$^2$ flasks (A and B) of Vero cells were infected with 1:1000 dilution of previous passage harvest and incubated for 7 days. The MOI for each passage (0.0005–0.005) was measured by back titration of the diluted viral inoculums. Viral RNA extracted from P1, P5, and P10, was used to generate NGS libraries using the TruSeq RNA Library Preparation Kit v2 (Illumina) without initial poly-A mRNA isolation procedures (flavivirus genomes lack a poly-A tail). Viral RNA and indexed library material were quantitated using a 2100 Bioanalyzer (Agilent) and paired-end NGS sequencing was performed on the MiSeq sequencing system (Illumina). The CLC Genomics workbench v.12 (Qiagen) was used for NGS assembly and variant calling. Duplicate read removal and trimming of paired-end reads were selected to reduce disproportionally high coverage caused by PCR amplification during library generation. Coverage and count filters were set as follows: ignore positions with coverage >100,000,000, minimum coverage = 10, and minimum count = 2. The reads were mapped against the reference sequence of each chimeric cDNA construct, and low frequency variant detection with 1% required significance was used for variant calling.

**Viral replication and temperature sensitivity in cell cultures**. Virus growth kinetics were conducted in Vero, LLC-MK2, and C6/36 cells grown in 75-cm$^2$ flasks. Cells were infected with each virus at a MOI of 0.001 and incubated with DMEM without FBS (Vero cells), DMEM containing 2% FBS (LLC-MK2 cells), or YE-LAH medium containing 2% FBS (C6/36 cells) in 5% CO$_2$ at 37 °C (Vero and LLC-MK2 cells) or 28 °C (C6/36 cells). For assessment of temperature sensitivity, an additional set of infected LLC-MK2 flasks was incubated in parallel at 39 °C. Aliquots of each culture medium were collected on indicated days and stabilized with an equal volume of medium containing 35% FBS and stored at −80 °C until

plaque titration. Experiments were performed in 2–5 flasks (Vero), 2–3 flasks (C6/36 cells), or duplicate flasks (LLC-MK2 cells for each temperature).

**Mosquito infection**. Viruses were cultured in Vero cells to generate virus stocks harvested 6–7 days post-infection (same day for oral infection of mosquitoes). The fresh virus harvests (never frozen) were mixed 1:1 with defibrinated calf blood (Colorado Serum Company) to make infectious blood meals. To ensure the infectious doses of vaccine viruses were similar to those of wt virus controls, both undiluted and 1:10 diluted viral harvests were included in experiments. Only groups fed with a desirable and comparable dose determined by back titration were included for final analysis. The virus doses in the mosquito experiments were: a) *Ae. aegypti*: 6.3, 6.5, 6.4, 5.9, 6.0; b) *Ae. albopictus* Lake Charles: 7.5, 7.2, 7.2, 7.4, 6.6; c) *Ae. albopictus* ELG: 7.4, 6.5, 6.4, 6.4, 6.4 log$_{10}$ pfu/mL for ZIKV, D2/ZK-V4, -V5, DENV-2 16681 and D2 PDK-53- VV45R, respectively.

Four to six day-old sugar-starved *Ae. aegypti* (established from a 2012 Poza Rica, Mexico collection[54]), *Ae. albopictus* (established from a 1987 Lake Charles, LA collection[20]), and *Ae. albopictus* from a newly established F8 colony (from a 2018 Estero Llano Grande (ELG) State Park collection in Weslaco, TX), were offered infectious bloodmeals for 30–45 min utilizing a Hemotek system (Discovery Workshops, United Kingdom). Mosquitoes were cold-anesthetized for sorting of fully engorged mosquitoes and then incubated for 14 days at 28 °C with 80% humidity and access to 10% sucrose under 14-10 (light-dark) cycles. Surviving mosquitoes were anesthetized with triethylamine (Flynap; Carolina Biological Supply Company) prior to collection of body, leg and saliva samples for determination of viral midgut infection, dissemination and transmission rate, respectively, as described[19]. The limit of detection was 5 pfu/body or legs and 2 pfu/saliva sample.

**Neurovirulence studies in newborn CD-1 mice**. Timed-pregnant female CD-1 mice were purchased from Charles River Labs Inc. and monitored for birth at DVBD/CDC. Mice were housed in micro-isolator cages in an animal room controlled at 20–26 °C with 30–70% humidity and 14-10 (light-dark) cycles. Newborn mice (0- to 5-day-old) were first evaluated for their susceptibility to lethal i.c. challenge with 10$^3$ or 10$^4$ pfu (in a volume of 30 μl) of DENV-2 16681 or ZIKV PRVABC59 to establish a highly sensitive and consistent neurovirulence model (based on morbidity/mortality rate) for the wt viruses. D2 PDK-53-VV45R was also included as a vaccine virus control. Based on the results, 1-day-old mice were used to evaluate the chimeric D2/ZK viruses by i.c. inoculation with 10$^4$ pfu of virus (≥ 10/group). A wt virus group (ZIKV or DENV-2) was included in each experiment as a positive neurovirulent control (100% morbidity). Animals were weighed and monitored daily for 21 days for signs of disease (lack of weight gain, lethargy, or abnormal posture or movement). Animals showing clear signs of illness were humanely euthanized following the approved animal protocol and were counted as non-survivors.

**Immunogenicity and efficacy studies in AG129 mice**. An AG129 mouse colony was maintained in a pathogen-free facility at DVBD/CDC with periodic introduction of new breeding pairs purchased from Marshall BioResources. Housing conditions for the mice were identical to the CD-1 mouse study. Three- to five-week-old, mixed sex AG129 mice were immunized with 10$^3$ or 10$^4$ pfu of D2/ZK viruses (4–13/group) via intraperitoneal (i.p.) inoculation on day 0. A booster immunization was given to double-dose groups on day 42. Mice were anesthetized and bled by cheek puncture for NAb testing between days 40–42 and 68–72. Mice were challenged with 10$^4$ pfu of ZIKV PRVABC59 via i.p. inoculation at 10 weeks post-primary immunization. On day 3 post-challenge, mice were bled and viremia was determined by plaque titration. Mice were monitored daily for weight loss and signs of disease caused by ZIKV (rough fur, hunched posture, lethargy, hind-limb paralysis) up to 28 days and animals showing signs of illness were humanely euthanized.

**Immunogenicity and efficacy studies in non-human primates**. The NHP in vivo study was performed at Alpha Genesis, Inc. (Yemassee, SC), and samples were sent to DVBD/CDC (Fort Collins, CO) and Takeda Vaccines (Cambridge, MA) for analysis. Indian rhesus macaques were pre-screened for previous flavivirus exposure using a Luminex assay[6]. One- to three-year-old, mixed sex, flavivirus-negative macaques were immunized with 10$^4$ pfu of chimeric vaccine viruses (6/group) or PBS (placebo control; 4/group) via s.c. inoculation on study day 0. Blood was collected on days 0–7, 9, 11 and 14 post-immunization for vaccine vRNAmia analysis. D2/ZK-V5 immunized animals received a second immunization 3 months post-primary immunization and were additionally bled on days 0, 3, 5, 7, 9, and 11 post-boost for vaccine vRNAmia analysis (study days 91–102) by qRT-PCR. All animals were challenged s.c. with 10$^4$ pfu of ZIKV PRVABC59 on study day 186. Blood was collected on days 0–10 and 14 post-challenge for vRNAmia analysis (study days 186–196 and 200). Blood collected on study days 9, 11, 14, 30, 45, 60, 91, 115, 157, 186, 196, 214 and 246 was used for measuring NAbs.

**Reporter micro-focus reduction neutralization test (R-mFRNT)**. The R-mFRNT was performed as previously described[5]. Briefly, live reporter chimeric WN/ZIKV (R-WN/ZKV) containing prM-E of ZIKV and a ZsGreen reporter gene in the

WNV replication backbone (constructed in-house at DVBD/CDC) was used for the high throughput micro-neutralization assay. Two-fold serial dilution of heat-inactivated serum was mixed with 100–200 viral focus forming units (ffu) of R-WN/ZKV, incubated for 1 h at 37 °C, then inoculated (30 μL/well, in triplicate) into a Greiner Cellstar tissue culture 96-well black-wall, flat-bottom plate (VWR) containing just-confluent Vero cell monolayers. After adsorption for 1.5 h, 150 μL/well of Gibco Fluorobite DMEM (ThermoFisher) without FBS was added, followed by incubation at 37 °C/5% $CO_2$. Plates were placed into a Celigo Image Cytometer (Nexcelom Bioscience) for automated live cell imaging and ffu counting within 24–28 h after infection. Back titration of the input R-WN/ZK was conducted in the same experiment and used to generate a neutralization curve of each sample based on 4-parameter non-linear regression in GraphPad prism software. The 50% effective concentration ($EC_{50}$) was calculated from the neutralization curve to indicate a titer providing 50% virus neutralization.

**Reporter virus particle (RVP) microneutralization assay**. A Zika RVP assay was also conducted as described[55] at Takeda Vaccines, Inc to confirm results of CDC's R-mFRNT. Briefly, heat-inactivated serum was serially diluted and then mixed with Zika RVPs. The diluted serum and RVP mixture was then plated in duplicate in a 384-well assay plate and incubated for 1 h at 37 °C. Vero cells were added to each well and incubated at 37 °C for 72 h. Renilla-Glo substrate (Promega) was then added to the plate, incubated for 15 min at room temperature, and the plate was subsequently analyzed by a luminometer. The $EC_{50}$ was determined by a non-linear regression curve fit. The serum titers indicated the final dilution level of the serum in the entire mixture of serum/RVP/Vero culture medium, while R-mFRNT (similar to traditional plaque reduction neutralization assay, PRNT) calculates the serum dilution level from serum/virus mixture only. As a result, the LOD and endpoint titer outcomes of the RVP assay are typically higher than those of R-mFRNT or PRNT.

**Quantitative real-time RT-PCR**. Total RNA was extracted from 140 μL of each serum sample using the QIAamp viral RNA mini spin kit (Qiagen), eluted in 60 μL of elution buffer, and stored at −80 °C until use. Viral RNA was quantified by qRT-PCR in triplicate using the same primers and probe as described previously[56]. The reaction was conducted with the QuantiTect Probe RT-PCR Kit (Qiagen) using primers and probe at concentration of 400 nM and 200 nM, respectively, in a 25 μL reaction containing 5 μL of vRNA on a CFX-96 Touch Real-Time PCR system (BioRad). Cycling conditions were as follows: 50 °C for 20 min and 95 °C for 5 min, followed by 45 cycles of 95 °C for 15 s and 57 °C for 75 s. Virus RNA concentration was determined by interpolating sample results with a standard curve generated from an in vitro transcribed vRNA made from the pD2i/ZK-V cDNA clone. The standard RNA was quantified by the Quant-iT RiboGreen kit (Molecular Probes) using a TBE-380 Mini-fluorometer (Turner Biosystems) and also verified against a synthetic ZIKV RNA segment purchased from ATCC. Total RNA from a negative control (flavivirus naive NHP serum) and from a positive control (comprised of 4.4 log pfu/mL of ZIKV PRVABC59 spiked into normal NHP serum) was included in each assay. The limit of quantification (LOQ) was calculated to be 3.3 $\log_{10}$ copies/mL, with linearity ($R^2$; 0.996), precision (σ; 0.457) and efficiency (96.97%) by R software (v 4.0.1, R Core team 2020). To ensure consistency of RNA extraction efficiency among samples, we also conducted a qRT-PCR assay to measure the quantity of rhesus macaque exosomal C1GALT1C1L mRNA in samples as previously described[6]. Using the same kit and instrument described above, primers and probe were used at final concentrations of 375 nM and 187.5 nM, respectively, in a 20 μL reaction containing 5 μL of RNA. Cycling conditions were as follows: 50 °C for 20 min and 95 °C for 5 min, followed by 45 cycles of 95 °C for 15 s and 60 °C for 60 s.

**Statistics**. Data were analyzed in GraphPad Prism software (V6 and V8) and Microsoft 365 Excel. Statistical comparisons among groups were performed using unpaired, two-tailed t-tests (plaque size and mosquito studies) or one-way ANOVA with Tukey's test to correct multiple comparisons(animal studies). The LOQ analysis for the qRT-PCR assay was performed by a code[57] available in the public domain using R software (v 4.0.1, R Core team 2020).

**Reporting summary**. Further information on research design is available in the Nature Research Reporting Summary linked to this article.

**Disclaimer**. The findings and conclusions in this report are those of the authors and do not necessarily represent the official position of CDC.

## Data availability

All data generated and analyzed in this study are included in the paper, supplementary information, and source data file. Genomic sequences of the viruses used in this study are available at GenBank (https://www.ncbi.nlm.nih.gov/genbank) with the following accession codes: U87411 for DENV-2 16681; U87412.1 for DENV-2 PDK-53; KU321639 for ZIKV SPH2015; KU501215.1 for ZIKV PRVABC59. Source data are provided with this paper.

## Code availability

The code used for the qRT-PCR LOQ calculation has been made available by the US Geological Survey/Department of Interior and can be found at https://doi.org/10.5066/P9GT00GB.

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

## Acknowledgements
This study was partly supported by a cooperative research and development agreement (CRADA #D-33-16) between the DVBD/CDC and Takeda. WRB and HAG were sponsored by Takeda under the CRADA as full-time guest researchers at CDC during the study. We wish to acknowledge the following DVBD/CDC colleagues for their assistance: Jason Velez for Vero cell plates, Brandy Russell for mouse monoclonal 4G2 antibody, Sean Masters for mouse husbandry and mosquito colony maintenance, Jeremy Leder-mann for help in establishing the new Ae. albopictus ELG colony at CDC and NGS, Kalanthe Horiuchi for statistical analysis to define LOQ of qRT-PCR assay, and members of the DVBD/CDC Sequencer Core team. We also want to acknowledge the Alpha Genesis, Inc. staff for support in the NHP in vivo study. Ae. Aegypti (Poza Rica 2012 collection) and Ae. albopictus (ELG 2018 collection) were kind gifts from Drs. Greg Ebel and William Black, Colorado State University, Fort Collins, CO.

## Author contributions
C.Y.H. conceived the initial research project. C.Y.H., H.J.D., and J.A.L. proposed and established the CRADA collaboration and guided the overall project direction. W.R.B., H.A.G., and J.L.S. contributed equally to the study. Engineering and deriving chimeric viruses: J.L.S., H.A.G., and C.Y.H. In vitro characterization of the vaccine candidates: J.L.S., H.A.G., and W.R.B.; Sanger and NGS sequencing: J.L.S. and H.A.G.; Mouse studies: J.L.S., W.R.B., H.A.G., and C.Y.H. Mosquito studies: H.A.G., W.R.B., and J.L.S.; NHP study: G.Y., C.Y.H., and W.R.B. (study design), G.Y. (in vivo study lead and coordination), W.R.B. (R-mFRNT), H.A.G. (qRT-PCR), K.J.B. (RVP). All authors contributed in data analysis and interpretation in experiments they conducted. W.R.B. and C.Y.H. reviewed all data, and made figures and tables. WRB wrote the first draft of the manuscript in coordination with H.A.G. and J.L.S. C.Y.H. revised and finalized the manuscript with inputs from all authors. All authors approved the final manuscript.

## Competing interests
G.Y., K.J.B., H.J.D., J.A.L., W.R.B., and H.A.G. are employees of Takeda. H.A.G. and W.R.B. were contracted by Takeda as CDC guest researchers during the study. C.Y.H.'s laboratory received CRADA fund provided by Takeda to partly support the study. C.Y.H. is an inventor of CDC patent applications based on DENV-2 PDK-53 based chimeric flavivirus vaccines, including the D2/ZIKV and TDV vaccines (TDV is licensed to Takeda for commercial development). The remaining authors declare no competing interests.
