## [Peer Review File · Nature Communications]

Single dose of chimeric dengue-2/Zika vaccine candidate protects mice and non-human primates against Zika virusReviewers' Comments:

Reviewer #1:

Remarks to the Author:

Thank you for the opportunity to review "A single dose of chimeric dengue 2 PDK-53/Zika vaccine candidate protects mice and non-human primates against Zika virus". This study describes the in vitro development and preclinical testing of live attenuated Zika virus (ZIKV) vaccine candidates in mice and non-human primates. The authors developed chimeric dengue virus (DENV)-2/ZIKV vaccine candidates including D2/ZK-V4, D2/ZK-V5, and D2/ZK-V5-Pr that grew to high titer in WHO Vero RCB 10-87 cell lines absent FBS, conditions required for vaccine manufacturing for human use. Following serial Vero cell passage, as would be required for vaccine production, the D2/ZK-V4 and D2/ZK-V5 candidates displayed genetic stability across three previously established PDK-53 attenuation determinants, although D2/ZK-V5-Pr did not and was thus excluded from subsequent in vivo evaluations. D2/ZK-V4 and D2/ZK-V5 displayed attenuated neurovirulence in 1-day-old CD-1 mice, attenuated virulence in IFN alpha, beta, and gamma receptor knockout AG129 mice, and inability to transmit via the main ZIKV urban mosquito vectors *Aedes (Ae.) aegypti* and *Ae. albopictus*. D2/ZK-V4 and D2/ZK-V5 candidates were found to be immunogenic and protective against wild-type ZIKV infection in AG129 mice and in Indian-origin rhesus macaques. The authors conclude that these results support further clinical development of the D2/ZK-V4 and D2/ZK-V5 vaccine candidates. Overall, the manuscript is well-written and the data appear valid and are clearly presented. One concern is that D2/ZK-V4 and D2/ZK-V5 viruses displayed multiple single nucleotide variants (SNVs) following serial passage, independent of three previously established PDK-53 attenuation determinants, which may have functional consequences of undetermined significance. This finding requires further discussion particularly in references to the authors conclusion that the D2/ZK-V4 and D2/ZK-V5 candidates are ready for human trials. An additional consideration is that the D2/ZK-V4 and D2/ZK-V5 candidates do not appear to have been evaluated for ability to induce antibody dependent enhancement (ADE), a relevant parameter for flavivirus vaccine design. This omission requires discussion. Please see my few additional comments below.

Specific Comments:

Page 9: The authors observe that the D2/ZK-V4, D2/ZK-V5, and D2/ZK -V5-Pr all replicated in C6/36 cells in the 4.2-5.1 log₁₀ pfu/ml range although neither D2/ZK-V4 and -V5 viruses successfully infected mosquitos. To what degree does viral titer in C6/36 cells predict infectivity in mosquitos?

P9: The authors indicate that small plaques, temperature sensitivity, restricted growth in C6/36 cells, low transmissibility in major *Aedes* mosquito vectors, and attenuation in neonatal mice are all previously defined in vitro and in vivo markers of DENV-2 PDK-53 vaccine attenuation. To what degree were each of these analyses carried out on passage 5 and 10 isolates of D2/ZK-V4 and -V5 viruses which displayed multiple SNV? If these analyses were not carried out on passage 5 and 10 isolates, please discuss what the implications might be for interpreting the attenuation phenotype of these highly passaged isolates.

P11. 1-day old CD-1 mice inoculated with D2/ZK-V4 and -V5 viruses all survived. Were D2/ZK-V4 and -V5 viruses successfully isolated from these mice to confirm productive infection? Were viral kinetics and weight changes across comparator groups evaluated in these experiments?

Reviewer #2:

Remarks to the Author:

In this study, Baldwin et al developed chimeric DENV-2/ZIKV vaccine candidates using the well-characterized and clinically proven dengue virus serotype-2 (DENV-2) PDK-53 vaccine backbone.

These candidates retained attenuated phenotypes of the PDK-53 vaccine virus such as attenuation of neurovirulence and virulence for mice and lack of transmissibility in the main mosquito vectors. Also, they provided results that showed a single DENV-2/ZIKV dose provided protection against ZIKV challenge in mice and rhesus macaques. The authors concluded that the live chimeric attenuated ZIKV vaccine candidates were safe, immunogenic and effective at preventing ZIKV infection in multiple animal models.

Overall, despite of a set of reported live attenuated ZIKV vaccines candidates, this study provided new chimeric ZIKV vaccine candidates for further development, and the majority of data well supported the claims of authors.

A major concern of the reviewer with this study is regarding the genetic stability of these vaccine candidates. As showed in Table 1 and Fig. 2, a large number of mutations, some of which resulted in AA change, appeared after passage 10 in both V4 and V5 candidates irrespective of passage lineages. These AA mutations may influence the phenotypic characteristics of vaccines candidates. The authors are suggested to further verify the attenuated phenotypes of these passaged viruses including plaque phenotype, temperature sensitivity as well as reduced virulence for mice.

Another weakness of this study is the authors didn't show the protection efficacy in pregnant mouse model.

Reviewer #1:

Thank you for the opportunity to review “A single dose of chimeric dengue 2 PDK-53/Zika vaccine candidate protects mice and non-human primates against Zika virus”. This study describes the in vitro development and preclinical testing of live attenuated Zika virus (ZIKV) vaccine candidates in mice and non-human primates. The authors developed chimeric dengue virus (DENV)-2/ZIKV vaccine candidates including D2/ZK-V4, D2/ZK-V5, and D2/ZK-V5-Pr that grew to high titer in WHO Vero RCB 10-87 cell lines absent FBS, conditions required for vaccine manufacturing for human use. Following serial Vero cell passage, as would be required for vaccine production, the D2/ZK-V4 and D2/ZK-V5 candidates displayed genetic stability across three previously established PDK-53 attenuation determinants, although D2/ZK-V5-Pr did not and was thus excluded from subsequent in vivo evaluations. D2/ZK-V4 and D2/ZK-V5 displayed attenuated neurovirulence in 1-day-old CD-1 mice, attenuated virulence in IFN alpha, beta, and gamma receptor knockout AG129 mice, and inability to transmit via the main ZIKV urban mosquito vectors *Aedes (Ae.) aegypti* and *Ae. albopictus*. D2/ZK-V4 and D2/ZK-V5 candidates were found to be immunogenic and protective against wild-type ZIKV infection in AG129 mice and in Indian-origin rhesus macaques. The authors conclude that these results support further clinical development of the D2/ZK-V4 and D2/ZK-V5 vaccine candidates. Overall, the manuscript is well-written and the data appear valid and are clearly presented.

1. One concern is that D2/ZK-V4 and D2/ZK-V5 viruses displayed multiple single nucleotide variants (SNVs) following serial passage, independent of three previously established PDK-53 attenuation determinants, which may have functional consequences of undetermined significance. This finding requires further discussion particularly in references to the authors conclusion that the D2/ZK-V4 and D2/ZK-V5 candidates are ready for human trials. More specifically, on P9, the reviewer comments: The authors indicate that small plaques, temperature sensitivity, restricted growth in C6/36 cells, low transmissibility in major *Aedes* mosquito vectors, and attenuation in neonatal mice are all previously defined in vitro and in vivo markers of DENV-2 PDK-53 vaccine attenuation. To what degree were each of these analyses carried out on passage 5 and 10 isolates of D2/ZK-V4 and -V5 viruses which displayed multiple SNV? If these analyses were not carried out on passage 5 and 10 isolates, please discuss what the implications might be for interpreting the attenuation phenotype of these highly passaged isolates.

We agree with the reviewer’s comment that introduction of SNVs during serial cell passages could result in functional consequences of undetermined significance and recognize that this is not evaluated in the current pre-clinical development study. Our goal in the current study is to evaluate genetic stability levels of the engineered vaccine viruses to select the candidates with suitable genetic stability for further clinical development. To be clear, the vaccine candidate virus culture stocks generated for the current pre-clinical study would not be appropriate for human clinical trials and so we have included additional discussion (lines 415-426 in clean version) to clarify that while the vaccine candidate viruses appear to be genetically stable, careful genotypic and phenotypic characterization of GMP-manufactured

clinical seeds would be required prior to human clinical trials (similar strategy as described in Ref #17).

- Ref#17: Huang, C. Y. et al. Genetic and phenotypic characterization of manufacturing seeds for a tetravalent dengue vaccine (DENVax). PLoS Negl Trop Dis 7, e2243, doi:10.1371/journal.pntd.0002243 (2013).

2. An additional consideration is that the D2/ZK-V4 and D2/ZK-V5 candidates do not appear to have been evaluated for ability to induce antibody dependent enhancement (ADE), a relevant parameter for flavivirus vaccine design. This omission requires discussion. We agree that ADE concern is important and have added discussion (line 398-412) and citations regarding the potential ADE risk of subsequent DENV infection following ZIKV vaccination, including potential strategies to mitigate such risk. Regarding the comment to conduct an ADE animal study of the vaccine candidates, we respectively argue that such a study is outside of the scope of the current study and cannot be conducted properly at this stage based on the following arguments:

Different Ab response profiles among humans and animals to different flaviviruses have been demonstrated and can directly affect ADE potential (cross-reactive Abs are generally less neutralizing with higher potential for ADE). For example, although ZIKV-infected subjects appear to generate significant Abs to virus-specific E-DIII (Ravichandran et al., 2019), it has been shown that DENV- or WNV-infected humans develop a dominant Ab response to the highly cross-reactive E-DI/DII (Slon Campos et al., 2018). In contrast, the mouse Ab response to flaviviruses predominantly target E-DIII. Therefore, an animal study to evaluate ADE potential of the vaccine should be conducted by passive transfer of human Abs elicited by vaccine candidates (available only after human clinical trials) to an appropriate animal model.

In addition, we consider most ADE animal models reported for flaviviruses to have significant limitations, and establishing a more accurate ADE model for flaviviruses is still needed before we can properly evaluate ADE potential of human Ab elicited by the ZIKV vaccine candidates. For example, passive transfer of human Ab (hAb) in a mouse ADE model has shown that pre-existing DENV and WNV hAb could enhance ZIKV infection (Bardina et al., 2017). However, years of epidemiologic and cohort studies so far have been able to demonstrate only certain DENV serotypes (especially DENV-2) exposures following a prior heterotypic DENV and/or ZIKV infection is positively correlated with enhancement of infection and disease severity (added Ref# 51). Importantly, in contrast to observations in multiple animal studies, human cohort studies have indicated that pre-existing DENV Ab protects against and does not enhance subsequent ZIKV infection in humans (added Ref

#50-51). The discordance between the mouse model and humans could be partly due to the significant differences in the Fc receptor types between murine and human cells (Bruhns, 2012), as Fc receptors are critical for binding the virus-Ab complex during ADE of infection.

Finally, ZIKV vaccine given with or after a tetravalent DENV vaccine may mitigate the risk of ADE in following DENV infection. Furthermore, D2/ZK LAV made on DENV-2 backbone could elicit DENV-2 cellular immunity and anti-NS1 antibodies (Ref#48 and added #52), both of which may decrease the risk of ADE outcomes in subsequent DENV infection. Overall, we think ADE risk of the D2/ZIKV LAV should be evaluated with immunized human samples during the clinical development stage in an animal model that can accurately mimic clinical human ADE outcomes. We believe the added paragraph in discussion should adequately address the reviewer's comment.

References:

- Ravichandran S, et al., Differential human antibody repertoires following Zika infection and the implications for sero diagnostics and disease outcome. *Nat Commun.* 2019 Apr 26;10(1):1943.
 - Slon Campos, J.L., Mongkolsapaya, J. & Screaton, G.R. The immune response against flaviviruses. *Nat Immunol* 19, 1189–1198 (2018).
 - Bardina SV, et al. Enhancement of Zika virus pathogenesis by preexisting ant flavivirus immunity. *Science.* 2017 Apr 14;356(6334):175-180.
 - Added Ref#50: Katzelnick, L. C et al. Zika virus infection enhances future risk of severe dengue disease. *Science* 369, 1123-1128, doi:10.1126/science.abb6143 (2020).
 - Added Ref#51: Katzelnick, L. C., Bos, S. & Harris, E. Protective and enhancing interactions among dengue viruses 1-4 and Zika virus. *Curr Opin Virol* 43, 59-70, doi:10.1016/j.coviro.2020.08.006 (2020).
 - Bruhns, P. Properties of mouse and human IgG receptors and their contribution to disease models. *Blood* (2012) 119 (24): 5640–5649
 - Ref# 48: Waickman, A. T. et al. Assessing the Diversity and Stability of Cellular Immunity Generated in Response to the Candidate Live-Attenuated Dengue Virus Vaccine TAK-003. *Front Immunol* 10, 1778, doi:10.3389/fimmu.2019.01778 (2019).
 - Added Ref# 52: Sharma, M. et al. Magnitude and Functionality of the NS1-Specific Antibody Response Elicited by a Live-Attenuated Tetravalent Dengue Vaccine Candidate. *The Journal of Infectious Diseases* 221, 867-877, doi:10.1093/infdis/jiz081 (2019).
3. Page 9: The authors observe that the D2/ZK-V4, D2/ZK-V5, and D2/ZK -V5-Pr all replicated in C6/36 cells in the 4.2-5.1 log₁₀ pfu/ml range although neither D2/ZK-V4 and -V5 viruses successfully infected mosquitos. To what degree does viral titer in C6/36 cells predict infectivity in mosquitos?

The C6/36 cell line is derived from *Aedes albopictus* larvae and is highly susceptible to most flaviviruses, including DENV and ZIKV. As the C6/36 cell line is only a single mosquito cell type and does not represent other mosquito cell types or the multiple infectivity barriers that the virus is required to overcome for productive infection in whole mosquitoes, replication efficiency in C6/36 cell culture cannot necessarily be used as a reliable indicator for mosquito competency of the virus. However, we have previously established a positive correlation of defective replication in C6/36 cells with poor infectivity in whole mosquitoes, specifically for the D2 PDK-53 virus, and have demonstrated that such dampened replication was attributable to the same PDK-53-specific attenuation determinants (Ref #11 and added #33). As shown in Fig 3, wt DENV, ZIKV, and all D2/ZKV-P viruses replicated to over 7-8 log₁₀ pfu/ml in C6/36 cells, while peak titers of the D2/ZK-V viruses (4-5 log₁₀ pfu/ml) were about 1,000-10,000 fold lower, exhibiting replication kinetics similar to the D2 PDK-53 vaccine backbone. In response to the comment, we have revised the manuscript to clarify that a positive correlation of the of D2 PDK-53 vaccine attenuation determinants between C6/36 cells and whole mosquitoes has been established and that the decreased replication efficiency (about 1000-fold lower) of the vaccine viruses in C6/36 cells relative to their parental counterparts has thus far reliably predicted attenuation in whole mosquitoes (line 346-353)

- Ref#11: Butrapet, S. et al. Attenuation markers of a candidate dengue type 2 vaccine virus, strain 16681 (PDK-53), are defined by mutations in the 5' noncoding region and nonstructural proteins 1 and 3. *J Virol* 74, 3011-3019, doi:10.1128/jvi.74.7.3011-3019.2000 (2000).
- Added Ref#33: Brault, A. et al. Replication of the primary dog kidney-53 dengue 2 virus vaccine candidate in *Aedes aegypti* is modulated by a mutation in the 5' untranslated region and amino acid substitutions in nonstructural proteins 1 and 3. *Vector Borne Zoonotic Dis* 11, 683-689, doi:10.1089/vbz.2010.0150 (2011).

4. P11. 1-day old CD-1 mice inoculated with D2/ZK-V4 and -V5 viruses all survived. Were D2/ZK-V4 and -V5 viruses successfully isolated from these mice to confirm productive infection? Were viral kinetics and weight changes across comparator groups evaluated in these experiments?

We have previously differentiated the mouse neurovirulence phenotypes of wt DENV-16681 virus and its derived PDK-53 vaccine virus (and associated PDK-53-based chimeric flavivirus vaccine candidates) by intracranial (i.c.) challenge of newborn ICR or Swiss mice utilizing a mortality endpoint (Ref #8-11, #16-17). Similar suckling mouse i.c. challenge models have also been widely used in other flavivirus LAV evaluation for decades, including a YFV-17D and JEV vaccine study showing high concordance between the neurovirulence profiles in suckling mice (mortality rate) and rhesus monkeys (histopathological brain lesion

scores) suggesting the neurovirulence mortality rate of the suckling mouse model could substitute for the monkey neurovirulence model as an appropriate safety test for neurovirulence of flavivirus vaccines (line 358-363 and added Ref #34-35).

This neurovirulence suckling mouse model was typically established to ensure consistently high morbidity rates caused by wt virus challenge. Because strain and age of the mice as well as virus dose can significantly affect the mortality rate, an appropriate model has to be established whenever we change the strain of the mice or tested new wt virus for such study. During the study, our previously established CDC in-house ICR or Swiss mouse colonies became unavailable. Therefore, we re-established a model based on a commercially available CD-1 strain for both wt ZIKV and DENV-2.

In response to this reviewer's comment, we included additional data (line 222-226 and Supplementary Fig 2) in establishing the neurovirulence mouse model. We observed that only 0-2 day old newborn mice consistently (100%) succumbed to lethal challenge by wt DENV-2 or ZIKV, while the majority of 4-5 day old mice were resistant to virus challenge. No virus isolation from mouse brain was conducted during the study, as it would require termination of mice for brain harvest at certain time points prior to flavivirus illness onset. Viremia cannot be assessed either, as the neonatal mice were too small to collect sufficient blood sample for virus isolation. However, mouse weights were measured daily for the study and we have added these data in new Fig 5b and line 230-234 as suggested by the reviewer. As expected, continued weight gain similar to the PBS group was observed among the chimeric virus groups, whereas average weight of mice challenged with either wt DENV or ZIKV showed minimum weight gain about 0-2 days before succumbing to disease. Furthermore, the i.c challenge procedure was conducted by a single investigator with >30 years of experience in the procedure with newborn mice, and all virus inocula were back titrated to ensure that the appropriate virus dose was administered to each mouse group. In addition, a lethal challenge group (wt DENV-2 16681 or wt ZIKV) was included in each experiment as a positive neurovirulent control to validate the success in i.c. inoculations. Overall, these data support our conclusion that the chimeric D2/ZIKV viruses are highly attenuated in the mouse neurovirulence model. Please see Fig 5b, Supplementary Fig 2, line 222-226, 230-234, 358-363, and 560-566 for all additional data, results, discussion, and methods related to this newborn mouse neurovirulence model.

- **Added Ref#34:** Jennings, A. Analysis of a yellow fever virus isolated from a fatal case of vaccine-associated human encephalitis. *J Infect Dis* 169, 512-518, doi:10.1093/infdis/169.3.512 (1994).
- **Added Ref#35:** Monath, T. P. et al., Single mutation in the flavivirus envelope protein hinge region increases neurovirulence for mice and monkeys but decreases viscerotropism for monkeys: relevance to development and safety testing of live,

attenuated vaccines. *Journal of virology*, 76, 1932–1943, doi:10.1128/jvi.76.4.1932-1943 (2002).

Reviewer #2:

In this study, Baldwin et al developed chimeric DENV-2/ZIKV vaccine candidates using the well-characterized and clinically proven dengue virus serotype-2 (DENV-2) PDK-53 vaccine backbone. These candidates retained attenuated phenotypes of the PDK-53 vaccine virus such as attenuation of neurovirulence and virulence for mice and lack of transmissibility in the main mosquito vectors. Also, they provided results that showed a single DENV-2/ZIKV dose provided protection against ZIKV challenge in mice and rhesus macaques. The authors concluded that the live chimeric attenuated ZIKV vaccine candidates were safe, immunogenic and effective at preventing ZIKV infection in multiple animal models.

Overall, despite of a set of reported live attenuated ZIKV vaccines candidates, this study provided new chimeric ZIKV vaccine candidates for further development, and the majority of data well supported the claims of authors.

1. A major concern of the reviewer with this study is regarding the genetic stability of these vaccine candidates. As showed in Table 1 and Fig. 2, a large number of mutations, some of which resulted in AA change, appeared after passage 10 in both V4 and V5 candidates irrespective of passage lineages. These AA mutations may influence the phenotypic characteristics of vaccines candidates. The authors are suggested to further verify the attenuated phenotypes of these passaged viruses including plaque phenotype, temperature sensitivity as well as reduced virulence for mice.

Same response to viewer #1-1 comment. We have included additional discussion (line 415-426) to clarify that careful genetic and phenotypic characterization of GMP-manufactured clinical seeds would be required prior to human clinical trials.

2. Another weakness of this study is the authors didn't show the protection efficacy in pregnant mouse model.

The studies in a pregnant mouse model are underway and this has been disclosed in the discussion (line 423-425). We did not include this data in the current report, as it entails large amounts of data from multiple mouse studies including infectivity of the LAV in male and female reproductive organs and protective efficacy to pregnant dam and fetus against ZIKV sexual and vertical infection. The inclusion would greatly increase the volume of the manuscript beyond the word, citation, and figure/table limits.

Reviewers' Comments:

Reviewer #1:

Remarks to the Author:

This reviewer thanks the authors for their thorough responses and edits. All of my concerns have been appropriately addressed